# Low Compute Unlearning via Sparse Representations

**Vedant Shah**  *vedantshah2012@gmail.com*
*Mila, Université de Montréal*

**Frederik Träuble**
*MPI, Tübingen*

**Ashish Malik**
*University of Oregon*

**Hugo Larochelle**
*Mila, Université de Montréal*

**Michael Mozer**
*University of Colorado, Boulder*

**Sanjeev Arora**
*Princeton University*

**Yoshua Bengio**
*Mila, Université de Montréal*

**Anirudh Goyal**  *anirudhg9119@gmail.com*
*Mila*

**Reviewed on OpenReview:** *https://openreview.net/forum?id=GyKXzmk43s*

## Abstract

Machine *unlearning*, which involves erasing knowledge about a *forget set* from a trained model, can prove to be costly and infeasible using existing techniques. We propose a low-compute unlearning technique based on a discrete representational bottleneck. We show that the proposed technique efficiently unlearns the forget set and incurs negligible damage to the model's performance on the rest of the data set. We evaluate the proposed technique on the problem of *class unlearning* using four datasets: CIFAR-10, CIFAR-100, LACUNA-100 and ImageNet-1k. We compare the proposed technique to SCRUB, a state-of-the-art approach which uses knowledge distillation for unlearning. Across all four datasets, the proposed technique performs as well as, if not better than SCRUB while incurring almost no computational cost.

## 1 Introduction

Machine Unlearning (Cao & Yang, 2015; Nguyen et al., 2022; Zhang et al., 2023; Xu et al., 2023; Kurmanji et al., 2023; Warnecke et al., 2021) may be defined as the problem of removing the influence of a subset of the data on which a model has been trained. Unlearning can be an essential component in addressing several problems encountered in deploying deep-learning approaches in practical scenarios. Neural networks such as Large Language Models (LLMs), trained on massive amounts of commonly available data, can exhibit harmful behaviors in the form of generating misinformation, demonstrating harmful biases, or other undesirable characteristics. A major culprit behind these behaviors is the presence of biased or corrupted instances in the training data of these models. To ensure safe model deployment, it is necessary to remove these instances. Another reason to remove instances and make a model behave as if it had not been trained on certain data

is concerns about data privacy and the right of end users to expunge their data (Mantelero, 2013; Dang, 2021). For example, an individual might want their data removed from a face recognition system that was trained on their faces such that it is no longer able to identify them. Several regulations are being put in place in order to safeguard the "right to be forgotten" (Pardau, 2018; Magdziarczyk, 2019). All the above problems can be addressed by unlearning the specific subset of the training data which gives rise to the harmful behavior of the model in the former cases and an individual's private data in the latter cases. Apart from these concerns, unlearning can also serve other purposes such as removing outdated data from a model to free up network capacity for more recent or relevant data. With increasing concerns about AI safety and the increasing ubiquity of deep learning models in real-world applications, the problem of unlearning is of critical importance.

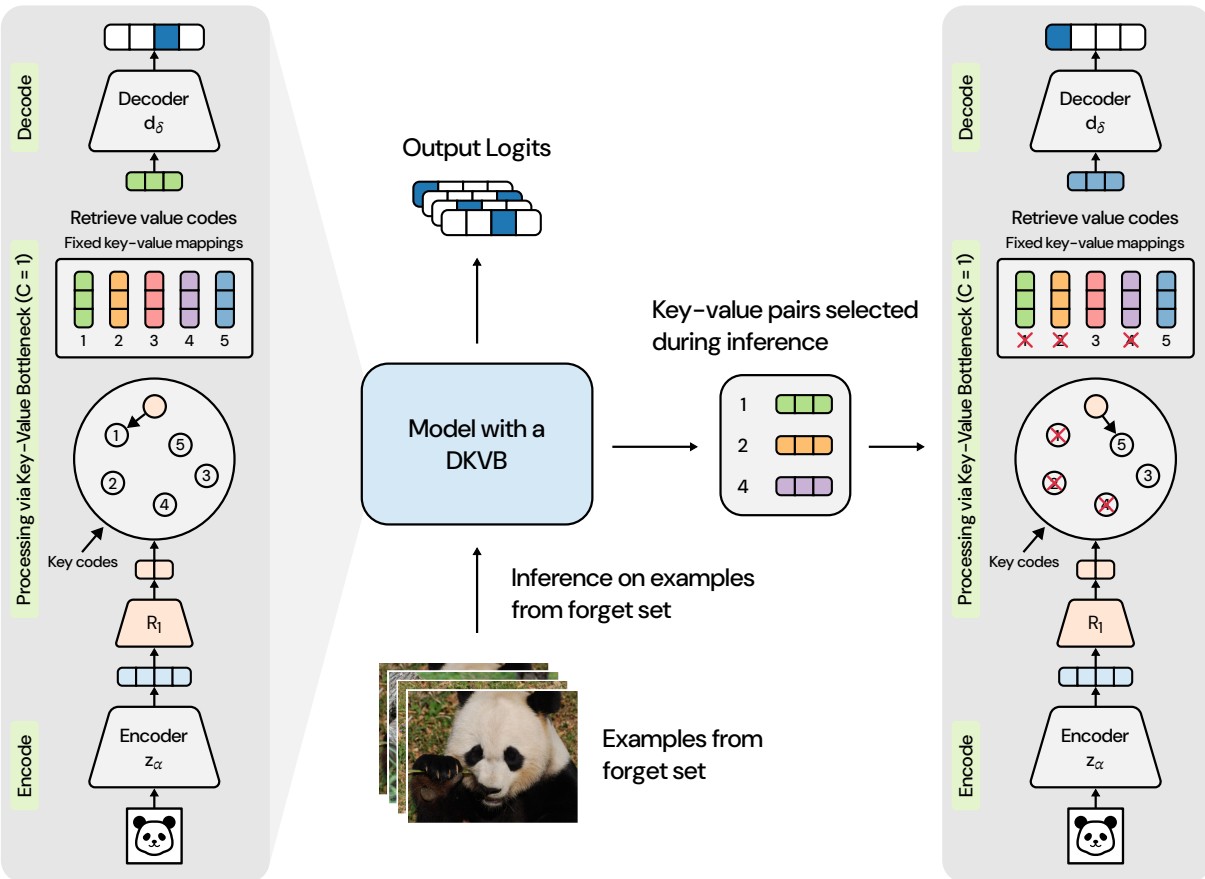

Figure 1: A summary of the proposed unlearning approach. **Left**: The structure of a key-value bottleneck. The encoder is frozen and pre-trained and $R_1$ is a random projection matrix. The values corresponding to the selected keys are retrieved to be used by the decoder. The gradient is backpropagated through the decoder into the values during training. The figure depicts the case with 1 codebook in the DKVB. However, in practice we use multiple codebooks. **Center:** Examples from the forget set are passed through the trained model and the key-value pairs selected during the forward pass are recorded. **Right:** The recorded key-value pairs are then masked from the bottleneck. As a result, the key selection is redirected to other keys, with non-informative corresponding values leading to other predictions.

The main challenge in unlearning is maintaining the performance of the model on the data that needs to be retained, called the *retain set*, while unlearning the *forget set*. The naive way to ensure that a model has no information about the forget set is to train from scratch on the retain set. Unlearning techniques aim to achieve the same goal but at a much lower computational cost compared to full retraining. Unlearning

in a pretrained network is difficult, especially in densely connected neural networks, since the value of one parameter may affect the output for all the input examples given to the neural network. A possible solution is to fine tune the model we wish to unlearn only on the retain set. While this would ensure that the performance of the model on the retain set is maintained, it can be computationally infeasible in practice. Other more effective solutions include retraining the model on the training data with a negative gradient for the forget set (Golatkar et al., 2020a; Kurmanji et al., 2023), or using knowledge-distillation-based training objectives to capture information about the retain set while filtering out information about the forget set (Kurmanji et al., 2023; Chundawat et al., 2023). Nevertheless, all of these approaches require some form of substantial additional compute in order to facilitate unlearning. Moreover, some of the existing approaches additionally require access to the original training data to facilitate unlearning, which may not be possible in many practical applications, e.g., for a model in production which is being trained online on an incoming data stream. The use of large models is becoming more popular and prevalent with the advent of general purpose transformer models. The requirement for additional compute can quickly become impractical in the context of these large models, especially in cases where a model is deployed and needs to be redeployed as quickly as possible after making the necessary changes.

In this article, we argue that specific kinds of *discrete neural information bottlenecks* are highly suited for very efficient and specific unlearning. Neural information bottlenecks have emerged as useful components in neural network architectures, providing numerous benefits such as improving out-of-distribution (OOD) generalization capabilities and robustness to noisy data (Goyal et al., 2021; Jaegle et al., 2021; Liu et al., 2021; 2023), facilitating large scale unsupervised pre-training and generative modeling (Esser et al., 2021; Oord et al., 2017), and more recently, helping in continual learning (Träuble et al., 2023). In particular, we build upon **Discrete Key-Value Bottleneck** (DKVB) proposed in Träuble et al. (2023). DKVB induces sparse representations in the form of key-value pairs which are trained in a *localized and context-dependent manner*. Since these representations are sparse, we hypothesize that it is possible to remove the information about a subset of the training data without damaging the information about the rest of the data—the primary desiderata for a useful unlearning method. Moreover, since the representations are discrete, this may be achieved without requiring any additional compute in the form of retraining or fine tuning, by directly intervening on individual representations.

We investigate the above-mentioned idea of low compute unlearning in the Discrete Key-Value Bottleneck. Specifically, we focus on the problem of *class unlearning* in multi-class classification tasks, where the aim is to remove information about a specific class, called the *forget class*, from a trained model. We use the term *retain classes* to refer to the classes other than the forget class that are present in the training data. More specifically, we wish to remove the *influence of the forget class* on the model. We measure this influence using the performance of the model on held-out test datasets corresponding to the forget class and the retain classes.

We propose two approaches for compute efficient unlearning in DKVB - *Unlearning via Examples* and *Unlearning via Activations*. We show that the proposed methods achieve unlearning of the forget class while incurring negligible damage to the model's performance on the retain classes. We compare the proposed methods to SCRUB (Kurmanji et al., 2023), a recent state-of-the-art approach that requires additional compute to unlearn, on four datasets: CIFAR-10, CIFAR-100, LACUNA-100 and ImageNet-1k. The novelty of our work lies in the largely under-explored idea of using a model architecture with *inherent sparse representations.*

## 2 Related Work

The problem of unlearning has been studied in different forms for over two decades. Early works such as Tsai et al. (2014), Cauwenberghs & Poggio (2000) and Duan et al. (2007) study the problem of *decremental learning* in linear models, where a small number of samples need to be removed from a model. Ginart et al. (2019) considers unlearning as a problem of deleting individual data points from a model. They give a probabilistic definition, formalize the notion of efficient data deletion, and propose two deletion efficient learning algorithms. Guo et al. (2019) introduces *certified removal* - a theoretical guarantee of indistinguishability between a model from which data was removed and a model that never saw the data. Izzo et al. (2021) distinguishes between exact unlearning and approximate unlearning and proposes a compute-efficient approximate data

deletion method, and a new metric for evaluating data deletion from these models. Golatkar et al. (2020a) and Kurmanji et al. (2023) cast unlearning into an information theoretic framework. Golatkar et al. (2020b) proposes Neural Tangent Kernel (NTK) (Jacot et al., 2018) theory-based approximation of the weights of the unlearned network. Multiple works also delve into the more philosophical, ethical, and legal aspects of unlearning and the "right to be forgotten" (Kwak et al., 2017; Villaronga et al., 2018). Chundawat et al. (2023) and Tarun et al. (2023) learn error minimization and error maximization-based noise matrices which are used to finetune the trained model in order to do unlearning. Chundawat et al. (2023) further uses a generator that generates pseudo data points for unlearning in order to operate in a data-free regime. Recent works have also explored unlearning in the context of Large Language Models. Liu et al. (2024); Mekala et al. (2024)

Kurmanji et al. (2023) introduces SCRUB, a knowledge distillation-based unlearning method. SCRUB considers the original model as a teacher model and trains a student model to obey the teacher model on the retain set and disobey it on the forget set. This is done by computing the KL Divergence between the output distributions of the two models and training the student model to maximize it on the forget set (called a *max-step*) and minimize it on the retain set (called a *min-step*). The student model is simultaneously also optimized for minimizing the task loss on the retain set. The training consists of *mstep max-steps*. The *max-steps* and *min-steps* are executed alternatively. Chen et al. (2023), similarly to us, focuses on class unlearning rather than unlearning specific instances in the data. Unlearning is done by destroying the decision boundary of the forget class. The authors propose two boundary shift methods termed as *Boundary Shrink* and *Boundary Expanding*.

**Unlearning and Model Sparsity.** Jia et al. (2023) and Mehta et al. (2022) investigate unlearning in context of model sparsity. Jia et al. (2023) leverages the Lottery Ticket Hypothesis, Frankle & Carbin (2018) leverages using parameter pruning on a trained dense model to identify the token subnetwork. They observe that applying standard unlearning approaches to a sparsified networks is better as compared to doing unlearning directly on the dense network. Mehta et al. (2022) identifies the Markovian Blanket of parameters corresponding to the examples to be unlearnt and updates those parameters. Fan et al. (2024) identifies the most *salient* weights with respect to the forget set and unlearns by finetuning only the salient weights with random labels. These approaches can be seen as applying sparse unlearning updates to the network instead of full finetuning. We point out that sparsity is a critical dimension that determines the effectiveness of unlearning: extremely sparse representations make unlearning trivial, whereas fully distributed representations intertwine knowledge in a way that makes compute-efficient unlearning a serious challenge. Previous methods studying sparsity in the context of unlearning such as Jia et al. (2023) and Mehta et al. (2022) propose the use of pruning techniques to first sparsify the network. These approaches start with dense trained models and leverage sparsity for unlearning. In contrast, we propose using sparsity as an in-built inductive bias in the model during the initial training which makes the model suitable for unlearning involving minimal compute requirements. On the other hand, Jia et al. (2023) sparsify the model after it has been trained. Mehta et al. (2022) involves sparse updates to the model parameters as discussed previously. However, these sparse updates are utilized during unlearning as opposed to during training of the original model in the proposed approach. We identify a sweet spot on the continuum between local and distributed learning that allows for both, compute-efficient unlearning and simultaneously obtaining the same generalization performance.

Xu et al. (2023) introduced a taxonomy that categorizes existing research on unlearning based on different approaches and aims. In this classification, our methods fall within the *Model Pruning* category by means of disabling specific (key, value) pairs within the bottleneck. Although one could argue that our methods lean towards a *weak unlearning* strategy–given the pre-trained backbone might retain some information about the forget set–our approach deviate from the strict definition of *weak unlearning* as outlined by Xu et al. (2023). As an example, when considering a non-parametric decoder, our methods affect intermediate rather than final model activations. Zhao et al. (2024) empirically shows that unlearning difficulty grows with entanglement between the embeddings of the forget set and the retain set. This analysis reinforces our design choice. A DKVB ensures minimal entanglement between the representations of the forget and retain sets.

Similar to the proposed approaches, Foster et al. (2024) and Schoepf et al. (2024) propose approaches for finetuning-free unlearning. The proposed approaches involve performing one backward pass to compute Fisher-information ratios, followed by algebraically scaling the handful of weights most specialised to the

forget set. While most of the approaches discussed above improve upon the naive and intractable baseline of retraining on the retain set, most of them (except (Foster et al., 2024) and Schoepf et al. (2024) require a substantial amount of additional computation in the form of optimizing an objective function for unlearning. This additional compute requirement can quickly become infeasible whenever large models are involved. The approach proposed in this work, on the other hand, requires negligible computation for unlearning. Any computation that may be required is in the form of running inference on the forget set. Further, most of the existing approaches for unlearning are optimized for models with continuous representations. We propose the first approach for unlearning in models with discrete representations. The proposed approaches show that discrete representations while providing several other advantages (such as better generalization), can also be leveraged for effective and compute efficient unlearning.

## 3 Background and Notations

**Unlearning**: Let $\mathcal{D}_{train} = \{x_i, y_i\}_{i=1}^{N}$ be a training dataset and $\mathcal{D}_{test}$ be the corresponding test dataset. In our experiments, we consider the setting of class unlearning, wherein we aim to unlearn a class $c$ from a model trained with a multiclass classification objective on $\mathcal{D}_{train}$. $c$ is called the *forget class* or the *forget set*. Given $c$, we obtain $\mathcal{D}_{train}^{forget} \subset \mathcal{D}_{train}$ such that $\mathcal{D}_{train}^{forget} = \{(x, y) \in \mathcal{D}_{train} | y = c\}$. The complement of $\mathcal{D}_{train}^{forget}$ is $\mathcal{D}_{train}^{retain}$, i.e., subset of $\mathcal{D}_{train}$ that we wish to retain. Thus $\mathcal{D}_{train}^{retain} \cup \mathcal{D}_{train}^{forget} = \mathcal{D}_{train}$. Similarly, from $\mathcal{D}_{test}$, we have $\mathcal{D}_{test}^{forget} = \{(x, y) \in \mathcal{D}_{test} | y = c\}$ and its complement $\mathcal{D}_{test}^{retain}$. We refer to $\mathcal{D}_{train}^{retain}$ and $\mathcal{D}_{test}^{retain}$ as the retain set training and test data; and $\mathcal{D}_{train}^{forget}$ and $\mathcal{D}_{test}^{forget}$ as the forget set training and test data, respectively.

**Discrete Key-Value Bottleneck**: A discrete key-value bottleneck (DKVB) (Träuble et al., 2023) consists of a discrete set of coupled key-value codes. The bottleneck contains $C$ codebooks with each codebook containing $M$ key-value pairs. Models with DKVB use a pre-trained and frozen encoder to encode the input into a continuous representation. This input representation is then projected into $C$ lower dimension heads and each head is quantized to the $top - k$ nearest keys in the corresponding codebook. The values corresponding to the selected keys are averaged, and used for the downstream task. The keys in the codebooks are frozen and initialized to cover the input data manifold whereas the values are learnable. The mapping between the keys and values is non-parametric and frozen. Thus, the gradient is not propagated between the values and keys during training of the model. Since the values are retrieved and updated sparsely, and all the components except the value codes and the decoder are frozen, DKVB stores information in the form of input-dependent, sparse and localized representations (i.e., the value codes). These inductive biases allow the framework to exhibit improved generalization under distribution shifts during training, as shown empirically as well as theoretically in Träuble et al. (2023). Figure 1 (Left) shows an overview of a model with a DKVB where $C = 1$, $M = 5$ and top-$k = 1$. While typically limited to multi-class image classification settings, Diera et al. (2024) discusses the application of DKVB in the context of encoder-only language models.

## 4 Unlearning via Sparse Representations

**Learning a Discrete Key Value Bottleneck.** A Discrete Key Value Bottleneck (DKVB) model is first trained on the given dataset using the standard negative log-likelihood (cross-entropy loss) training objective for multi-class classification. We use a non-parametric average pooling decoder and test the proposed approaches on two pretrained backbones: 1.) a CLIP (Radford et al., 2021) pre-trained ViT-B/32 (Dosovitskiy et al., 2020) and 2.) a ResNet-50 pretrained on ImageNet in a supervised fashion. Then we proceed to unlearn a specific subset of data from these models. Before training with the classification objective, we do a *key initialization* for the DKVB models on the same dataset.

**Key Initialization in DKVB models.** After being forward propagated through the pre-trained encoder, the representations of the input are mapped to the top-k closest keys in the information bottleneck. The mapping between keys and values in the discrete key-value bottleneck is non-parametric and frozen. As a result, there is no gradient (back)propagation from the values to the keys, and hence the keys are not modified during training. Thus, it becomes essential for the keys to be initialized before learning the values and decoder, such that they broadly cover the feature space of the encoder. This initialization helps the

model represent different concepts effectively. As in Träuble et al. (2023), we use exponential moving average (EMA) updates (Oord et al., 2017; Razavi et al., 2019) to initialize the keys of the DKVB models. The key-initialization is done on the same train dataset $\mathcal{D}_{train}$ which we want to train the model on. The key initializations depend solely on the input encodings of the backbone and hence do not require access to any labeled data.

**Inference for Unlearning.** We propose to achieve unlearning in DKVB models by excluding key-value pairs from the bottleneck such that they cannot be selected again. Numerically, this masking is done by setting the quantization distance of the selected keys to 'infinity'. Figure 1 (center and right column) shows an overview of the proposed methods. More specifically, we experiment with two methods, *Unlearning via Activations* and *Unlearning via Examples*, described as follows.

**Unlearning via Examples.** In this method, we analyze the effect of unlearning a subset of $N_e$ examples belonging to the forget set. $N_e$ examples are randomly sampled from the forget set training data ($\mathcal{D}_{train}^{forget}$) and are input into the model having a DKVB. All key-value pairs that are selected during forward propagation across the $N_e$ examples are flagged. These key-value pairs are then masked out from the bottleneck. Technically, this approach requires access to the original training data corresponding to the forget class. However, it is also possible to carry out this procedure with a proxy dataset that has been sampled from a distribution close enough to that of the forget set. For more discussion on this, refer to Appendix A.1.

**Unlearning via Activations.** In this second method, we analyze the effect on the quality of unlearning by deactivating different numbers of key-value pairs corresponding to the forget set. We refer to the key-value pairs that have been selected as inputs to the decoder as *activations*. The entire forget set is forward-propagated through the DKVB model and all the key-value pairs selected across all examples of the forget class are recorded. Next, we mask the top-$N_a$ most frequently selected key-value pairs from the bottleneck. The requirement of accessing the original training data for this method can be avoided by caching all the activations corresponding to the forget set during the last epoch of training. Further, similar to the previous case, unlearning via activations may also be performed given access to data that has been sampled from a distribution close enough to the distribution of the forget set.

Both approaches are two different ways of achieving a common objective: to exclude a subset of activations corresponding to the forget set. However, using one approach over the other may be more practical or even necessary, depending on the task at hand. In both the above approaches, we do not do any form of retraining or fine-tuning. The only computation which may be necessary is incurred during the inference stage for recording the key-value pairs which have been utilized for the forget set. Hence both approaches require negligible additional compute. Moreover, the requirement of access to original training data of the forget class can also be circumvented under appropriate assumptions, making the proposed approaches *zero-shot unlearning methods*.

## 5 Experiments and Results

The goal of our experiments is two-fold. First, we validate that proposed methods of the *Unlearning via Activations* and *Unlearning via Examples* in models with a DKVB (Section 5.2), and show that the proposed methods are competitive with the baselines (Section 5.2.1) in unlearning the forget class while incurring minimal damage to the performance of the models on the retain class. Second, we compare the compute efficiency of the proposed methods against that of the baselines. More specifically, we report the number of floating-point operations (FLOPs) required during the procedure of unlearning. Before presenting these results, we describe our experimental setup (Section 5.3). Before presenting these results, we describe our experimental setup.

### 5.1 Experimental Setup

**Benchmark datasets** We validate the proposed methods using experiments across four base datasets: CIFAR-10 with 10 distinct classes, CIFAR-100 (Krizhevsky et al., 2009) with 100 distinct classes, LACUNA-100 (Golatkar et al., 2020a) with 100 distinct classes and ImageNet-1k (Russakovsky et al., 2015) with 1000 distinct classes. LACUNA-100 is derived from VGG-Faces (Cao et al., 2018) by sampling 100 different

celebrities and sampling 500 images per celebrity, out of which 400 are used as training data and the rest are used as test images.

**Models** On the aforementioned three datasets we study the following types of model architectures:

(a) **Backbone + Discrete Key-Value Bottleneck (Ours)**: Overall, this architecture consists of three components: 1) the frozen pre-trained backbone 2) the Discrete Key-Value Bottleneck (DKVB) and 3) a decoder, as shown in Figure 1. For the DKVB, we use 256 codebooks, with 4096 key-value pairs per codebook (approximately 1M pairs overall) as in Träuble et al. (2023).

(b) **Backbone + Linear Layer (Baseline)**: As a baseline, we replace the Discrete Key Value bottleneck and the decoder in the above model architecture with a linear layer. Thus, the two components of this model are 1) a frozen pre-trained backbone and 2) a linear layer. This model will be used for all the baseline methods. We replace the DKVB with a linear layer in the baseline since we observe that existing unlearning approaches are designed for continuous representation architectures and thus perform better on them.

In each model, we use a pre-trained frozen CLIP (Radford et al., 2021) ViT-B/32 and ImageNet supervised pre-trained ResNet-50 as our encoder backbones. We refer the reader to the appendix for additional implementation details.

**Training the Base Models** We then train both model architectures on the full training sets of each dataset. Since the backbone is frozen, for the baseline models, only the weights of the linear layer are tuned during initial training (and later unlearning). Since we use only one linear layer, we do not do any pre-training (beyond the backbone), unlike in previous works (Kurmanji et al., 2023; Golatkar et al., 2020a;b). Table 3 shows the performance of these trained models on the train and test splits of the complete datasets. Starting from these base models trained on the full datasets, we will validate the ability to unlearn previously learned knowledge.

**Key-Initializations**. To remain consistent with the literature on DKVB (Träuble et al., 2023), we apply EMA-based key-initializations to the DKVB models in all cases. The key-initializations lead to a clean class-wise separation of the keys in the DKVB as discussed in (Träuble et al., 2023). Consequently, they enable high-quality unlearning by making sure that different classes have minimal number of shared keys. We empirically explore the effect of key-initializations on the quality of unlearning in Appendix A.4.

**Unlearning** We aim to make the problem of unlearning as challenging as possible. In order to bias the comparison against the proposed methods we select the forget class to be the class best learnt by the models with the DKVB in each case (see Appendix A.3). The intuition behind this experimental design choice is that the class best learnt by the model would be associated with the largest number of representations in the DKVB (leading to high values of $N_a$ and $N_e$ for unlearning), thus increasing the likelihood of it sharing representations with other classes. Thus, unlearning these classes would lead to a higher drop in the retain set performance as compared to unlearning other classes. Nevertheless, we experimentally demonstrate that the proposed approaches perform competitively even when the forget set is randomly chosen in Appendix A.9.

**Objective & Metrics** We report our results on the test data of retain classes and forget class, i.e. $\mathcal{D}_{test}^{retain}$ and $\mathcal{D}_{test}^{forget}$. Typically in unlearning, the performance of the model on training examples from the forget set is studied. We study unlearning in a more broader sense and aim to study whether the model unlearns general "information" about the class, i.e., given a forget set, is the model's performance on all similar data points (and not just those belonging to the forget class training data) drops; i.e. whether the unlearning generalizes. Thus, we report the performance of the models on the forget class evaluation set. Note that this practice would not be valid in case of instance specific unlearning, wherein the aim is to unlearn specific examples of the training data from the model.

Although the proposed approaches facilitate unlearning up to arbitrary extents, in our experiments, we aim to achieve *complete unlearning* - achieving minimal accuracy on the forget set while incurring minimal damage to the performance on the retain set. We do this in order to stress test the proposed approaches. The greater the extent of unlearning, the larger the extent of damage incurred on the performance of the model on the retain classes.

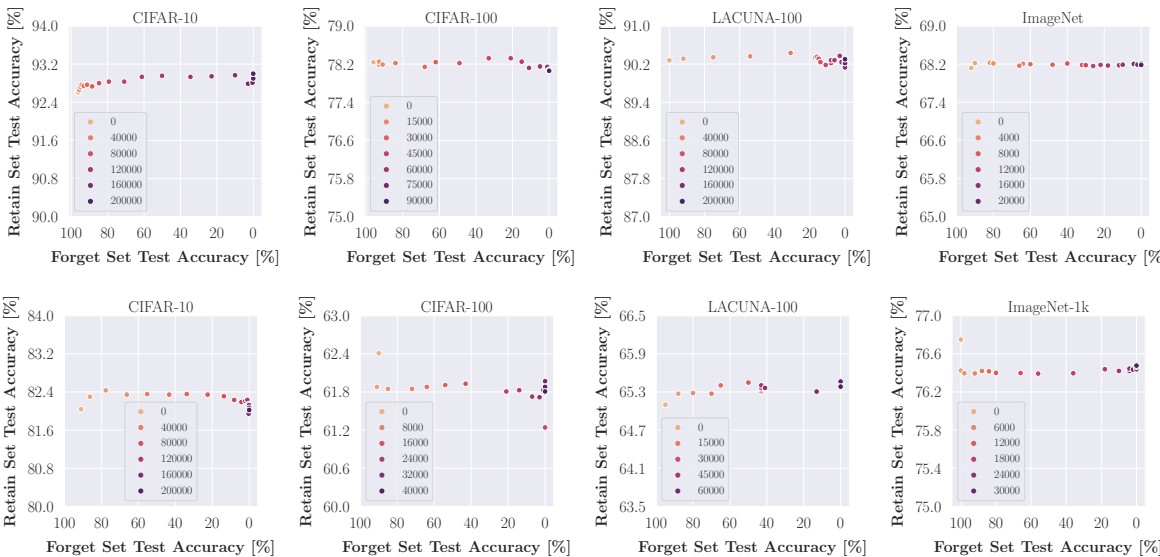

Figure 2: **Unlearning via Activations.** Performance on the retain set test data vs. Performance on the forget set test data across various datasets for (a) CLIP pretrained ViT/B-32 in the **top row** (b) ImageNet pretrained ResNet-50 backbones in the **bottom row** as the value of $N_a$ is increased which is indicated by the color of the markers. The relative performance on the retain set test data as compared to the original models increases after unlearning in the case of CIFAR-10 and ImageNet-1k and drops for CIFAR-100 and LACUNA-100 in the case of ViT/B-32 and increases for all four datasets in the case of ResNet-50 (see Table 1).

While achieving complete unlearning may not always be desirable, such as in the case of Membership Inference Attacks (MIAs) the proposed methods can be easily extended to defend against MIAs (We refer to Appendix A.11 for further discussion on MIAs and the proposed methods). We report mean values across 5 random seeds in all cases.

For comparing the compute efficiency of different approaches, we report the approximate FLOPs (Floating Point Operations) required for the procedure of unlearning. The total number of FLOPs are calculated as *number of FLOPs required during the forward passes + number of FLOPs required during the backward passes.* We use the `fvcore`[1] library for computing the number FLOPs required during the forward passes. FLOPs required using backward passes are approximated as *number of operations used for gradient computations + number of operations used for weight updates*[2]. Since only the linear head weights are trainable, the number of computations required for calculating the gradients would be the same as the number of parameters in the linear layer. Further, since we use Adam optimizer, the number of operations required for the weight updates would be equal to 18 times the number of parameters.

To calculate the final number of FLOPs, we first calculate the FLOPs required for one example (one forward + backward pass) and then multiply them with the total number of examples and the total number of epochs. For SCRUB, the forward and backwards FLOPs are multiplied with different scalars depending on the *msteps* parameter.

## 5.2 Unlearning via the Discrete Key-Value Bottleneck

We will now discuss the results of unlearning via activations and examples, i.e. the two approaches proposed in Section 4 on all four benchmark datasets.

**Unlearning via Activations.** Unlearning via activations requires us to set the hyperparameter $N_a$, reflecting the top-$N_a$ most frequently activated key-value pairs which will be masked out after inference on the forget

---

[1]https://github.com/facebookresearch/fvcore/
[2]https://epochai.org/blog/backward-forward-FLOP-ratio

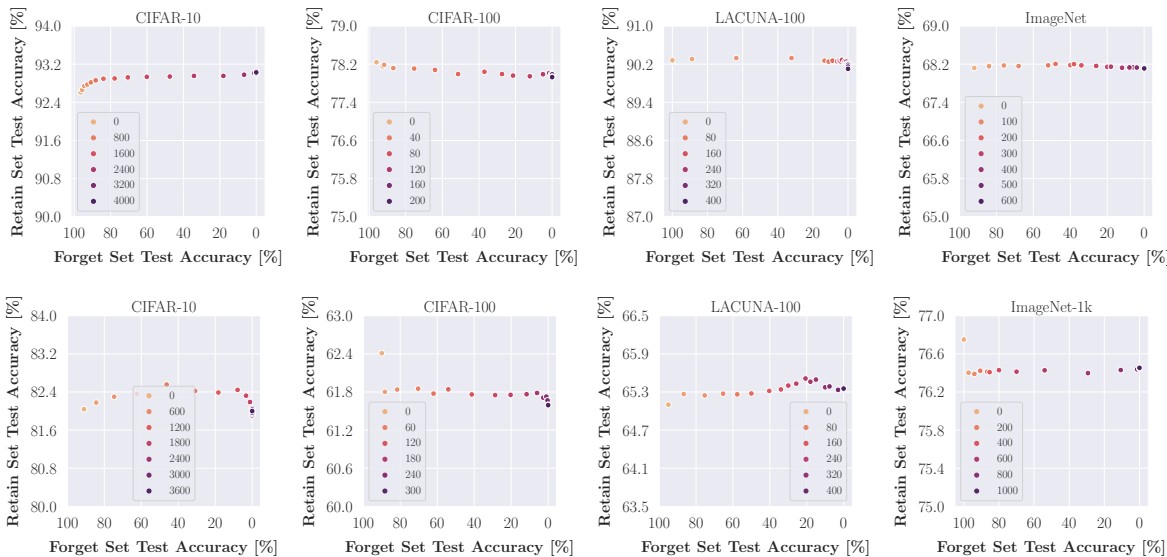

Figure 3: **Unlearning via Examples**. Performance on the retain set test data vs. Performance on the forget set test data across different datasets for (a) CLIP pretrained ViT/B-32 in the **top row** and (b) ImageNet pretrained ResNet-50 backbones in the **bottom row** as the value of $N_e$ is increased which is indicated by the color of the markers. The relative performance on the retain set test data as compared to the base model increases for CIFAR-10 and drops for all other datasets in the case of ViT/B-32, whereas it drops for CIFAR-10 and CIFAR-100 and increases for LACUNA-100 and ImageNet-1k in the case of ResNet-50 (see Table 1)

set. We therefore start by analyzing its role over a wide range of values to probe its choice and effect with $N_a = 0$ being the limit without any unlearning. Figure 2 summarizes the unlearning and effect of $N_a$ on the retain vs forget test set. In the case of CIFAR-10 and a ViT/B-32 backbone, the initial accuracies on the retain and forget test set are 92.61% and 96.50% respectively. As $N_a$ increases, the forget class test accuracy decreases, slowly for small $N_a$ and rapidly for larger $N_a$. The model reaches random accuracy (i.e. 10% for CIFAR-10) on the forget class test data at $N_a = 150000$. At this point the retain set test accuracy is 92.97%. The model unlearns the forget class completely between $N_a = 170,000$ (0.4%) and $N_a = 180,000$ (0%). That is, the forget class test accuracy of the DKVB model drops to 0% between these values of $N_a$. The exact value of $N_a$ at which the accuracy drops to zero, represents the minimum number of activations that need to be discarded in order to destroy all information about the forget class from the model. At this point, the retain set test accuracy is 92.94%, which is almost identical to the initial accuracy. Further increasing $N_a$ up to $N_a = 200,000$, i.e. about 20% of all key-value pairs, leads to an additional increase in retain set test accuracy to 93%. On the contrary, in the case of CIFAR-10 and a ResNet-50 backbone, the decrease in the forget set test accuracy is rapid for small $N_a$ and it slow for higher values of $N_a$. Complete unlearning in this case happens between $N_a = 160000$ (0.1%) and $N_a = 190000$ (0%). These differences in trends can be attributed to how the information is factorized among the representations. For eg. there is a steep decline in the forget set test accuracy between $N_a = 50000$ (41%) and $N_a = 55000$ (13%) in the case of LACUNA-100 and ResNet-50 backbone. This behavior may be attributed to the presence of high information but less frequently selected key-value pairs between the two values of $N_a$. Nevertheless, as can be seen from the equivalent analysis on the CIFAR-100, LACUNA-100 and ImageNet-1k models in Figure 2 and Figure 3, the same trend of maintaining the initial retain accuracy while minimizing the forget accuracy up to a minimum holds across all four datasets and both backbones validating its meaningful unlearning capability.

**Unlearning via Examples.** For the second method—unlearning via examples—$N_e$ examples are sampled randomly from the training data of the forget class, and subsequently used for unlearning by the mechanism described in Section 4. Similar to before, we aim to assess the effect on the choice of $N_e$ over a wide range for each dataset, including $N_e = 0$ being the limit without any unlearning. Figure 3 summarizes the unlearning and effect of $N_e$ on the retain vs. forget test set. We again begin by focusing on the results with CIFAR-10

Table 1: Comparison between the proposed methods and the baseline across CIFAR-10, CIFAR-100, LACUNA-100 and ImageNet-1k datasets and CLIP pretrained ViT/B-32 and ImageNet pretrained ResNet-50 backbones. We compare the **relative change in performance** on the retain and forget set test data relative to the originally trained models. The proposed methods are able to unlearn the forget sets completely in all cases while causing minimal changes in the performance of the models on the retain set test data.

| Backbone | Method | CIFAR-10 | | CIFAR-100 | | LACUNA-100 | | ImageNet-1k | |
| | | $\mathcal{D}_{test}^{retain}$ | $\mathcal{D}_{test}^{forget}$ | $\mathcal{D}_{test}^{retain}$ | $\mathcal{D}_{test}^{forget}$ | $\mathcal{D}_{test}^{retain}$ | $\mathcal{D}_{test}^{forget}$ | $\mathcal{D}_{test}^{retain}$ | $\mathcal{D}_{test}^{forget}$ |
|---|---|---|---|---|---|---|---|---|---|
| ViT/B-32 | DKVB via Activations (sec 5.2) | 0.36% | -100% | **-0.20%** | -100% | -0.17% | -100% | 0.15% | -100% |
| | DKVB via Examples (sec 5.2) | 0.45% | -100% | -0.36% | -100% | **-0.09%** | -100% | **-0.03%** | -100% |
| | Linear Layer + SCRUB | 1.62% | -100% | -0.91% | -100% | -1.10% | -100% | **7.31%** | -100% |
| | Linear Layer + Finetuning | **1.94%** | -100% | -1.91% | -98.33% | -2.21% | -100% | 0.88% | -100% |
| | Linear Layer + Retraining | 1.82% | -100% | -0.39% | -100% | -2.03% | -100% | 5.16% | -100% |
| | Linear Layer + NegGrad+ | 0.49% | -100% | -0.63% | -100% | -1.34% | -100% | 2.45% | -100% |
| ResNet-50 | DKVB via Activations (sec 5.2) | 0.04% | -100% | 0.26% | -100% | 0.21% | -100% | 0.04% | -100% |
| | DKVB via Examples (sec 5.2) | -0.07% | -100% | -0.34% | -100% | 0.17% | -100% | **0.04%** | -100% |
| | Linear Layer + SCRUB | -0.07% | -99.67% | -0.94% | -98.79% | -0.26% | -99.67% | **0.74%** | -100% |
| | Linear Layer + Finetuning | 0.48% | -100% | -0.46% | -99.99% | -2.96% | -100% | -2.25% | -100% |
| | Linear Layer + Retraining | **3.06%** | -100% | **1.76%** | -100% | 1.15% | -100% | -1.14% | -100% |
| | Linear Layer + NegGrad+ | 2.13% | -100% | -0.85% | -100% | **6.73%** | -100% | -0.85% | -100% |

and ViT/B-32 backbone. Here, the forget set $\mathcal{D}_{train}^{forget}$ contains 5000 examples. We start off with retain set and forget set test accuracies of 92.61% and 96.50% respectively. Similar to the previous approach – unlearning via activations – the test accuracy on the forget set decreases with increasing $N_e$. The accuracy on the retain test set, on the other hand, increases monotonically, although only slightly overall. The model achieves random accuracy on the forget class around $N_e = 2500$. The accuracy on retain set test data is at just under 93% at this transition. Finally, the accuracy on the forget set drops to 0% (i.e. complete unlearning) between $N_e = 3000$ and $N_e = 3400$ with a retain set test accuracy of just above 93% at $N_e = 3400$. Further increasing $N_e$ does not affect the retain set test accuracy notably. Similarly to the case of unlearning via activations, the forget set test performance decreases rapidly at first and then slowly with $N_e$ in the case of CIFAR-10 with a ResNet-50 backbone. The retain set test accuracy increases at first and then decreases, albeit marginally. An equivalent analysis on the CIFAR-100, Lacuna-100 and ImageNet-1k models in Figure 3 exhibit a similar behavior of successful minimization of the forget accuracy up to a minimum while roughly maintaining the retain set test accuracy, validating unlearning via examples as another option for unlearning using discrete key-value bottlenecks.

**Summary.** Both methods, *Unlearning via Activations* and *Unlearning via Examples*, successfully demonstrate unlearning of the forget class while having a negligible effect on the models' performance on the retain set. Importantly, this is achieved without any form of training, retraining, or fine-tuning as is usually required by other methods. The retain set test accuracy remains more or less constant for all four datasets except for a few minor fluctuations. This is a result of the fact that due to localized and context-dependent *sparse updates* during the initial training of the model, discrete key-representations corresponding to different classes in the dataset are well separated from each other, an important prerequisite discussed in Träuble et al. (2023). Hence, all the information about a class can be unlearned by forgetting only a subset of the forget class training data in the case of *Unlearning via Examples*, making it very data-efficient. While the aforementioned experiments are conducted in the context of unlearning a single class, Appendix A.8 further discusses the performance of the proposed approaches in multi-class unlearning scenarios.

### 5.2.1 Comparison with Baselines

We now compare the results of both the proposed methods, which require Backbone + DKVB models against several baseline methods, which are optimized for models without such a bottleneck. For this, we will use the Backbone + Linear Layer models described in 5.1. On these models, we run SCRUB (Kurmanji et al., 2023), *finetuning* - finetuning the model to be unlearnt on the retain set, *retraining* - training the model from scratch on the retain set only and NegGrad+ (Kurmanji et al., 2023) and compare the performance changes on the forget and retain classes against the performance changes after unlearning with the two proposed methods. Table 1 shows the comparison between the two previously reported methods and the baselines. We can see that one of the two proposed approaches always results in the least change in the performance of the base model on the retain classes, while at the same time achieving complete unlearning of the forget class. The baselines on the

Table 2: Comparison of FLOPs for various methods across CIFAR-10, CIFAR-100, LACUNA-100, and ImageNet-1k datasets using ViT/B-32 and ResNet-50 backbones. We report both forward and backward FLOPs for each method.

| Backbone | Method | CIFAR-10 | | CIFAR-100 | | LACUNA-100 | | ImageNet-1k | |
|---|---|---|---|---|---|---|---|---|---|
| | | Forward (TFLOPs) | Backward (GFLOPs) | Forward (TFLOPs) | Backward (GFLOPs) | Forward (TFLOPs) | Backward (GFLOPs) | Forward (TFLOPs) | Backward (GFLOPs) |
| ViT/B-32 | DKVB via Activations (sec 5.2) | 21.93 | **0** | 2.19 | **0** | 1.75 | **0** | 5.63 | **0** |
| | DKVB via Examples (sec 5.2) | **14.91** | **0** | **0.75** | **0** | **1.40** | **0** | **2.90** | **0** |
| | Linear Layer + SCRUB | 655.13 | 14.59 | 1316.83 | 293.30 | 527.60 | 117.51 | 39168.28 | 87230.98 |
| | Linear Layer + Finetuning | 196.54 | 4.38 | 6485.87 | 1444.61 | 5188.69 | 1155.69 | 11179.75 | 24898.21 |
| | Linear Layer + Retraining | 196.54 | 4.38 | 1080.98 | 240.77 | 864.78 | 192.61 | 5589.87 | 12449.11 |
| | Linear Layer + NegGrad+ | 393.08 | 8.76 | 1729.56 | 385.23 | 1383.65 | 308.18 | 11179.75 | 24898.21 |
| ResNet-50 | DKVB via Activations (sec 5.2) | 16.66 | **0** | 1.67 | **0** | 1.33 | **0** | 5.31 | **0** |
| | DKVB via Examples (sec 5.2) | **7.33** | **0** | **0.93** | **0** | **1.33** | **0** | **3.74** | **0** |
| | Linear Layer + SCRUB | 1498.74 | 87.55 | 1488.79 | 869.68 | 3697.00 | 2159.62 | 31873.28 | 299227.17 |
| | Linear Layer + Finetuning | 1049.12 | 61.29 | 1648.66 | 963.07 | 131.89 | 77.05 | 5304.25 | 49796.43 |
| | Linear Layer + Retraining | 659.47 | 385.23 | 1648.66 | 963.07 | 2242.18 | 1309.78 | 5304.25 | 49796.43 |
| | Linear Layer + NegGrad+ | 329.73 | 192.61 | 10608.49 | 99592.85 | 263.79 | 154.09 | 5304.25 | 49796.43 |

other hand, occasionally fail to achieve complete unlearning. Finally, it is important to re-emphasize that the proposed methods achieve the shown performance without requiring any additional gradient-based training for unlearning. In the case of baselines, we stop the unlearning procedure when the forget set is completely unlearned or the forget set test accuracy has converged with minimal damage to the performance on the retain set. Moreover, while we report results for the case of complete unlearning, the proposed methods can be easily used for achieving unlearning of the forget class to different extents by tuning the $N_a$ and $N_e$ hyperparameters. We refer to Appendix A.13.3 and A.13.2 for further training and implementation details.

## 5.3 Proposed Methods achieve Unlearning in a Compute Efficient Manner

In this section, we compare the proposed approaches against the baselines in terms of the amount of compute required in order to achieve complete unlearning. To facilitate this comparison, we report the number of FLOPs required for the unlearning procedure for each case. The FLOPs are calculated following the rules described in Section 5.1. Table 2 compares the FLOPs required for unlearning in each case.

We report the forward and backward FLOPs separately to highlight that the proposed approaches do not require any gradient based updates. Additionally, while the scale of backward FLOPs may seem insignificant against the forward FLOPs, it can easily blow up in cases where complex parametric decoders are used on top of the DKVB. In our experiments, the decoder is simply an average pooling layer. Nevertheless, we can see that the proposed approaches require significantly less forward FLOPs as compared to the baselines. This can be explained by the fact that the proposed approaches require only one forward pass through the models per example of the forget class training data, in order to cache the activations. The baseline methods on the other hand, require multiple forward passes, each corresponding to a single training epoch.

We also provide a runtime comparison of the proposed approaches with SCRUB in Appendix A.5, showing that the proposed approaches are at least $20\times$ more compute efficient than SCRUB, further demonstrating the efficacy of the proposed approaches.

## 6 Limitations and Future Work

The proposed methods inherit the limitations of the DKVB (Träuble et al., 2023) such as the reliance of DKVB on pre-trained encoders which can extract meaningful shared representations and trade-offs in downstream performance due to the use of an information bottleneck. Extensions to the model may involve training sparse representations inducing discrete bottleneck end-to-end. Further, in our experiments, we consider the setting of *multi-class classification* in a *supervised learning* setting where the forget set can be easily identified and isolated. However, this may not always be sufficient for a given task and more complicated approaches might be needed to identify the data that needs to be removed from the model. Scaling the proposed framework and evaluating its effectiveness in more complex scenarios such as generative modeling remains to be explored. While the methods introduced in this work are currently not designed for selective unlearning as outlined in Appendix A.11, there are various directions for future adaptations. These directions include enhancing the forget set isolation, and addressing limitations related to information retention, for instance by further fine-tuning on the encoder backbone.

Further, while we use the performance of the model on the test set of the forget class as a proxy for unlearning quality, it is important to note that it is not the only indicator of the unlearning quality. The forget class test accuracy could be deteriorated simply by attenuating the weights of the final linear layer corresponding to the

forget class, in the baseline models. However, it would be trivial to relearn the classifier head as compared to relearning information unlearned from a feature extractor. For complex tasks, the DKVB would typically be a part of the feature extractor. This distinction is not captured by the forget class test accuracy and is not reflected in our experimental setup due to the simplicity of the tasks considered. We leave extending the framework to more complex tasks for future work.

## 7 Conclusion

In this work, we proposed a new approach to unlearning that requires minimal computation in order to unlearn a subset of data. This approach is based on the use of a discrete architectural bottleneck which induces sparse representations. These sparse representations facilitate unlearning a subset of data from the model with minimal to no performance drop on the rest of the data. We focused on the setting of class unlearning and our experiments show that the proposed approach, while being at least $20\times$ compute efficient, performs competitively with or in some cases better than a state-of-the-art approach which requires additional compute to perform unlearning. Consequently, excising the activated key-value pairs from the model is a highly effective means of unlearning the forget set without disrupting the retain set.

## Acknowledgements

This research was enabled by compute resources provided by Mila (mila.quebec). VS would like to thank Aniket Didolkar and Moksh Jain for useful discussions, feedback on the initial versions of the paper and proof reading the paper. VS would also like to thank Nanda H Krishna and Moksh Jain for helping with the figures included in the paper and Thomas Jiralerspong for proof reading and giving feedback on the paper.

## Broader Impact Statement

This paper presents work whose goal is to advance the field of Machine Unlearning. Machine Unlearning has emerged as a very important problem with the recent advances in Artificial Intelligence. With the advent of large, compute intensive neural networks, out work can have significant societal impact by addressing the applications of machine unlearning such as privacy preservation and mitigating harmful among many others discussed in the paper.

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

## A  Appendix

### A.1  Unlearning in the absence of original training data

The main condition for unlearning the forget class in the bottleneck is that the keys which are closest to the encoder representations of the forget set examples in the forget class are being removed. One way to do this as described above, is by recording which keys get selected for the forget class examples and subsequently removing them from the bottleneck. However, in the absence of the forget class training data, the same could also be done by passing examples not directly belonging to the forget set but drawn from a distribution that is close enough to the forget set. This will result in approximately the same set of keys being selected as would have been if the examples belonged to the forget set.

### A.2  Initial Performances of the Models

We train both: the models with a DKVB and the baseline models (i.e. backbone + linear layer) to achieve similar performances on the test datasets in order to ensure a fair comparison. Note that, due to these the models are not necessarily trained to achieve the maximum possible performance on the datasets. Table 3 shows the initial performances of the originally trained models on different splits of the datasets.

### A.3  Deciding the Forget Class

We assume that this class should be the most difficult one for the model to forget. Figures 4(a) - 4(c) show the number of mis-classifications per class on the test data, for CIFAR-10, CIFAR-100 and LACUNA-100 for the ViT-B/32 backbone. For CIFAR-10, class #1 is the best-learned class with the lowest number of mis-classifications. Thus, we select class #1 as the forget class for the dataset. For CIFAR-100 class 58 is the

Table 3: Performance of the models on different sets of data after the initial training on the four datasets. We use two kinds of models: (a) models having a Discrete KV Bottleneck which are used for the proposed methods and (b) models where the DKVB and the decoder are replaced by a Linear Layer. These are used for the baseline. We wish to reduce the accuracy of these models on $D_{test}^{forget}$ to 0% while maintaining the accuracy on $\mathcal{D}_{test}^{retain}$.

(a) **Backbone + DKVB**

| | ViT-B/32 | | | | | | ResNet-50 | | | | | |
|---|---|---|---|---|---|---|---|---|---|---|---|---|
| **Dataset** | $\mathcal{D}_{train}$ | $\mathcal{D}_{train}^{retain}$ | $\mathcal{D}_{train}^{forget}$ | $\mathcal{D}_{test}$ | $\mathcal{D}_{test}^{retain}$ | $\mathcal{D}_{test}^{forget}$ | $\mathcal{D}_{train}$ | $\mathcal{D}_{train}^{retain}$ | $\mathcal{D}_{train}^{forget}$ | $\mathcal{D}_{test}$ | $\mathcal{D}_{test}^{retain}$ | $\mathcal{D}_{test}^{forget}$ |
| CIFAR-10 | 100% | 100% | 100% | 93.01% | 92.61% | 96.50% | 100% | 100% | 100% | 82.94% | 82.04% | 91.00% |
| CIFAR-100 | 99.98% | 99.98% | 100% | 78.43% | 78.24% | 96.00% | 99.98% | 99.98% | 100% | 62.11% | 61.81% | 92.00% |
| LACUNA-100 | 98.09% | 98.07% | 100% | 90.38% | 90.28% | 100% | 98.35% | 98.34% | 100% | 65.53% | 65.24% | 94.00% |
| ImageNet-1k | 99.53% | 99.53% | 100% | 68.24% | 68.22% | 92.00% | 99.37% | 99.36% | 100% | 76.44% | 76.41% | 100% |

(b) **Backbone + Linear Layer**

| | ViT-B/32 | | | | | | ResNet-50 | | | | | |
|---|---|---|---|---|---|---|---|---|---|---|---|---|
| **Dataset** | $\mathcal{D}_{train}$ | $\mathcal{D}_{train}^{retain}$ | $\mathcal{D}_{train}^{forget}$ | $\mathcal{D}_{test}$ | $\mathcal{D}_{test}^{retain}$ | $\mathcal{D}_{test}^{forget}$ | $\mathcal{D}_{train}$ | $\mathcal{D}_{train}^{retain}$ | $\mathcal{D}_{train}^{forget}$ | $\mathcal{D}_{test}$ | $\mathcal{D}_{test}^{retain}$ | $\mathcal{D}_{test}^{forget}$ |
| CIFAR-10 | 93.27% | 92.82% | 97.32% | 93.02% | 92.59% | 96.90% | 83.80% | 83.42% | 87.20% | 82.04% | 81.74% | 91.70% |
| CIFAR-100 | 86.73% | 86.61% | 99.00% | 78.53% | 78.35% | 96.00% | 78.02% | 77.82% | 98.20% | 62.69% | 62.41% | 90.00% |
| LACUNA-100 | 95.58% | 95.53% | 100% | 90.68% | 90.59% | 100% | 86.48% | 84.53% | 99.25% | 65.40% | 65.10% | 95.00% |
| ImageNet-1k | 73.13% | 73.11% | 96.15% | 68.29% | 68.26% | 92.00% | 97.44% | 97.43% | 100% | 76.77% | 76.75% | 100% |

best-learned class and for LACUNA-100, class 48 is one of the best-learned classes with zero mis-classifications. Hence, we select classes #58 and #48 as the forget classes for CIFAR-100 and LACUNA 100 respectively. We determine the forget class in all other cases using the same method. We use the same forget classes for experiments on the models with a linear layer in place of the DKVB (i.e., the baseline) as well. Table 4 shows the forget class for all the cases discussed in our experiments.

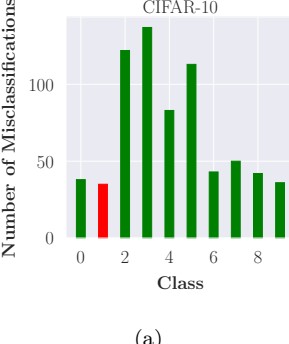
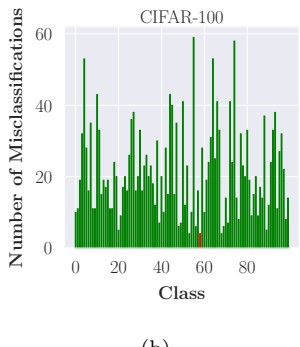
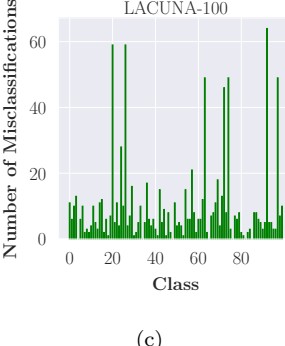

(a)  (b)  (c)

Figure 4: Number of mis-classifications per class for the test data. The red bars correspond to the class with the least number of mis-classifications (a) CIFAR-10: Class 1 has the least number of mis-classifications (b) CIFAR-100: Class 58 has the least number of mis-classifications (c) LACUNA-100: Classes 34, 48, 65, 76, 82 and 85 have 0 mis-classifications and hence, do not have a bar

Table 4: Forget classes for the different scenarios presented in the paper

| Forget Classes | CIFAR-10 | CIFAR-100 | LACUNA-100 | ImageNet-1k |
|---|---|---|---|---|
| ViT-B/32 | 1 | 58 | 48 | 1 |
| ResNet-50 | 1 | 94 | 34 | 9 |

## A.4 Importance of Key-Value Initializations in the DKVB

The effectiveness of the proposed approaches depends on the quality of key-initializations (i.e. how well the keys corresponding to different classes are separated), which also affects the overall performance of the DKVB model. EMA is used for key-initializations and thus plays a crucial role in the approaches. The quality of unlearning would deteriorate in the absence of key-initializations. We demonstrate this with an experiment where we compare the retain set accuracy of ViT-B/32 backbone based DKVB models trained with and without key-initializations when they undergo unlearning via activations for CIFAR-10, CIFAR-100 and LACUNA-100 and report the results in Table 5. We can see that the retain set accuracy incurs significant damage upon complete unlearning when the models are trained without key-initializations.

Table 5: Comparison of the effectiveness of Unlearning via Activations with and without EMA based key-initialization for ViT-B/32 backbone based models

| DKVB via Activations | CIFAR-10 | | CIFAR-100 | | LACUNA-100 | |
|---|---|---|---|---|---|---|
| | $\mathcal{D}_{\text{test}}^{\text{retain}}$ | $\mathcal{D}_{\text{test}}^{\text{forget}}$ | $\mathcal{D}_{\text{test}}^{\text{retain}}$ | $\mathcal{D}_{\text{test}}^{\text{forget}}$ | $\mathcal{D}_{\text{test}}^{\text{retain}}$ | $\mathcal{D}_{\text{test}}^{\text{forget}}$ |
| w/ Key-Init | 0.36% | -100% | -0.20% | -100% | -0.17% | -100% |
| w/o Key-Init | -2.96% | -100% | -13.36% | -100% | -9.46% | -100% |

In order to stay consistent with the literature on DKVB, we use EMA, exactly as used in (Träuble et al., 2023), using the same parameter values (i.e. gamma = 0.95). We discuss the implementation in detail in the Appendix A.12.1.

## A.5 Comparison of Runtimes

In this section, we present a comparison of runtimes of the proposed approaches against the baseline, SCRUB (Kurmanji et al., 2023) as a proxy for comparing the compute requirements of the two approaches. Table 6 presents the results of this comparison across all cases. We observe that in all the cases, the runtime of the baseline is multiple orders of magnitude greater than the runtime of the proposed approaches, with proposed approaches being at-least $20\times$ faster than the baseline.[3] Large runtimes for SCRUB are a result of the gradient-based parameter optimization required for the approach.

## A.6 Comparisons to fully trained baselines

We demonstrate perform equally well in comparison to baselines that are fully trained by training the baselines to completion on CIFAR-10 and CIFAR-100 for ResNet-50 backbones, and comparing unlearning on them via SCRUB with unlearning in the proposed approaches.

We first report the performances of the fully trained baselines and corresponding DKVB models in Table 7. For the baseline models, we run an extensive hyperparameter sweep and select models according to the best test performance across all the combinations.

Next, we run SCRUB (Kurmanji et al., 2023) on the fully trained baseline models and compare the performance with the proposed approaches in Table 8.

Clearly, unlearning on fully trained baselines is more difficult than on models trained to the same extent as the DKVB models.

## A.7 Unlearning in SCRUB

Figure 5 plots the retain class test accuracy vs forget class test accuracy for running SCRUB on a (CLIP pretrained and then finetuned on CIFAR-100) ViT-B/32 backbone in the case of CIFAR-100 (similar to Figures 2 and 3). The forget set accuracy drops to 0% after the first epochs. We run the unlearning procedure for 10 epochs, each epoch consisting of either one or two optimization steps, depending on the *msteps*

---

[3]Note that the runtimes for ImageNet-1k are low since we use pre-computed backbone embeddings to run these experiments.

Table 6: Comparison of runtimes (in **seconds**) between the proposed methods and the baseline on CIFAR-10, CIFAR-100, LACUNA-100 and ImageNet-1k. The proposed methods are at-least $20\times$ faster than the baseline across all cases. Between the two proposed methods, *Unlearning via Activations* is faster.

| Backbone | Method | CIFAR-10 | CIFAR-100 | LACUNA-100 | ImageNet-1k |
|---|---|---|---|---|---|
| ViT/B-32 | DKVB via Activations | **5.02** | **1.57** | **2.74** | **0.84** |
| | DKVB via Examples | 13.98 | 6.65 | 13.03 | 2.36 |
| | Linear Layer + SCRUB | 288.80 | 921.56 | 553.31 | 181.04 |
| ResNet-50 | DKVB via Activations | **4.88** | **0.84** | **1.50** | **1.02** |
| | DKVB via Examples | 12.26 | 2.39 | 2.20 | 4.30 |
| | Linear Layer + SCRUB | 1703.73 | 1902.89 | 4192.96 | 295.45 |

Table 7: Performance of the models on different sets of data after the initial training on the four datasets.

(a) **Backbone + DKVB**

| | ResNet-50 | | | | | |
|---|---|---|---|---|---|---|
| **Dataset** | $\mathcal{D}_{train}$ | $\mathcal{D}_{train}^{retain}$ | $\mathcal{D}_{train}^{forget}$ | $\mathcal{D}_{test}$ | $\mathcal{D}_{test}^{retain}$ | $\mathcal{D}_{test}^{forget}$ |
| CIFAR-10 | 100% | 100% | 100% | 82.94% | 82.04% | 91.00% |
| CIFAR-100 | 99.98% | 99.98% | 100% | 62.11% | 61.81% | 92.00% |

(b) **Backbone + Linear Layer (Fully Trained)**

| | ResNet-50 | | | | | |
|---|---|---|---|---|---|---|
| **Dataset** | $\mathcal{D}_{train}$ | $\mathcal{D}_{train}^{retain}$ | $\mathcal{D}_{train}^{forget}$ | $\mathcal{D}_{test}$ | $\mathcal{D}_{test}^{retain}$ | $\mathcal{D}_{test}^{forget}$ |
| CIFAR-10 | 86.68% | 85.96% | 93.08% | 84.28% | 83.50% | 91.3% |
| CIFAR-100 | 81.61% | 81.45% | 97.2% | 63.3% | 63.11% | 82.00% |

parameter. As explained in Section 5.2.1, we run SCRUB until the damage on the retain set test accuracy is minimal.

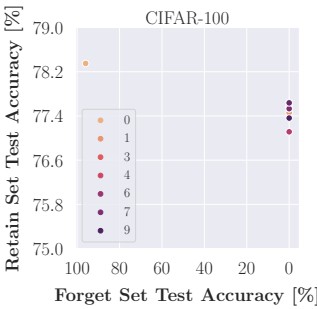

Figure 5: Retain Class Test Accuracy vs Forget Class Test Accuracy. The markers are color coded to represent the number of epochs.

Table 8: Comparison of the proposed approaches with unlearning via SCRUB on fully trained baseline models. F-T stands for fully trained baselines.

| | | CIFAR-10 | | CIFAR-100 | |
|---|---|---|---|---|---|
| Backbone | Method | $\mathcal{D}_{test}^{retain}$ | $\mathcal{D}_{test}^{forget}$ | $\mathcal{D}_{test}^{retain}$ | $\mathcal{D}_{test}^{forget}$ |
| ResNet-50 | DKVB via Activations (sec 5.2) | 0.04% | -100% | 0.26% | -100% |
| | DKVB via Examples (sec 5.2) | -0.07% | -100% | -0.34% | -100% |
| | Linear Layer + SCRUB | -0.07% | -99.67% | -0.94% | -98.79% |
| | Linear Layer + SCRUB (F-T) | -1.15% | -99.99% | -2.35% | -98.37% |

## A.8 Multi Class Unlearning

We attempt to investigate the effects of unlearning multiple classes at once by performing experiments on CIFAR-100 for both ViT/B-32 and ResNet-50 models. We unlearn upto 10 classes using both - Unlearning via Activations as well as Unlearning via Examples as well as one of the baselines - SCRUB (Kurmanji et al., 2023) and run each experiment for 5 seeds. The classes to be forgotten are chosen randomly for each seed. Figure 6 plts relative change in performance of the unlearnt model on the retain class (with respect to the original model) vs the number of classes unlearnt at approximately the point of complete unlearning.

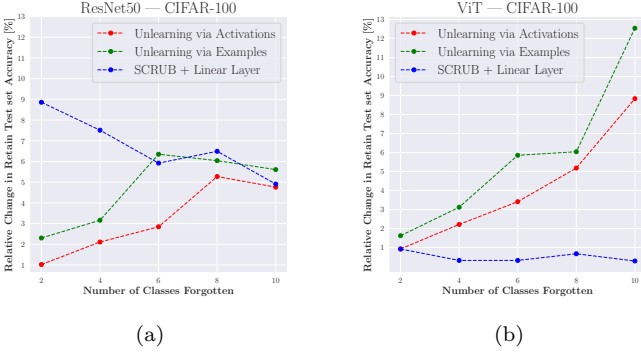

Figure 6: Multi Class Unlearning for CIFAR-100

We can see that Unlearning via Activations performs relatively better as compared to Unlearning via Examples. Further, SCRUB outperforms both the proposed methods significantly in the case of a ViT backbone, keeping the percentage change in the retain class test accuracy less than 1% in all cases. However, in the case of a ResNet-50 backbone, SCRUB surprisingly performs the worst for low number of unlearnt classes and competitively for higher number of unlearnt classes. The proposed methods perform comparatively better on the ResNet-50 backbone as compared to the ViT backbone. We can also clearly see that the relative error as compared to the original model on the retain set performance is higher than single class unlearning, as expected. Noticeably, the performance starts to significantly deteriorate when forgetting > 6 classes in the case of Unlearning via Examples, and > 8 examples in the case of Unlearning via Activations for both the backbones.

## A.9 Choosing the forget class randomly

To ensure that the effectiveness of the approach is not class specific, we perform experiments on CIFAR-100, where the class to be forgotten is randomly chosen, and compare the performance of the proposed approaches against SCRUB [2]. We run each experiment for 5 random seeds, wherein the forget class is randomly chosen for each seed. Rest of the experimental setup remains the same as described in Section 5 of the paper. We report the results in Table 9

Table 9: Selecting the forget class randomly for CIFAR-100

| Backbone | Method | CIFAR-100 | |
| --- | --- | --- | --- |
| | | $\mathcal{D}_{test}^{retain}$ | $\mathcal{D}_{test}^{forget}$ |
| ViT/B-32 | DKVB via Activations (sec 5.2) | -0.27 ± 0.07 % | -100% ± 0 % |
| | DKVB via Examples (sec 5.2) | -0.47 ± 0.03 % | -100 ± 0 % |
| | Linear Layer + SCRUB | -1.56 ± 0.35% | -100 ± 0% |
| ResNet-50 | DKVB via Activations (sec 5.2) | 0.58 ± 0.12 % | -100 ± 0 % |
| | DKVB via Examples (sec 5.2) | 0.28 ± 0.11 % | -100 ± 0 % |
| | Linear Layer + SCRUB | -1.43 ± 0.18% | -99.94 ± 0% |

Clearly, even with the forget class chosen randomly, the proposed approaches perform competitively to SCRUB. This demonstrated that the effectiveness of the proposed approaches is not class dependent.

### A.10 Unlearning beyond the compute free setting

We investigate the effect of using additional compute to the proposed methods. As shown previously, the proposed methods perform competitively to SCRUB. To the best of our knowledge, SCRUB is the most competitive and relevant unlearning approach. However, it has the inherent drawback of requiring compute for unlearning. Nevertheless, for a fair comparison, we additionally explore the implications of this additional compute for the proposed two methods for a ViT/B-32 backbone on CIFAR-10, CIFAR-100 and LACUNA-100. Specifically, we retrain the DKVB models after the (compute efficient) unlearning, on the training data of the retain set (i.e., $\mathcal{D}_{train}^{retain}$) for 10 epochs. For the baseline, we use the same experimental setting as in Section 5.2.1, except - we run it for 10 epochs instead of stopping when either the forget set has been completely unlearned or the performance has converged. Figure 7 and figure 8 highlight the effect of retraining of the proposed methods compared to SCRUB across multiple epochs, for all three datasets.

Retraining the unlearned models on the retain set does not affect their performance significantly. The performance of the baseline on the other hand increases after an initial drop in case of CIFAR-100 and LACUNA-100. The initial drop may be attributed to the damage to the retain set performance caused by the initial *max-steps*. The subsequent increase can be attributed to the fact that the SCRUB training objective also optimizes the task loss on the retain set. Thus, once the model unlearns the forget set, SCRUB shifts the model capacity towards better learning the retain set. For CIFAR-10 this results in the model performing better than the DKVB models on the retain set as the retain set test accuracy after unlearning is higher than the original model. However, the baseline is unable to recover its original performance for CIFAR-100 and LACUNA-100.

For the forget set, in all three cases, the baseline completely unlearns the forget set quickly within the first few epochs, as shown in Figure 8.

### A.11 Using the proposed Methods against Membership Inference Attacks

Depending on the application, complete unlearning of the forget set may not always be the final goal of unlearning. For several use cases such as removing information about corrupted data from the model or removing harmful biases exhibited by the model, maximal error on the forget set is desirable. However, for applications such as Differential Privacy, it is more desirable to achieve a forget set error that is similar to that of a model trained from scratch only on the retain set. Otherwise, it makes the unlearned model susceptible to Membership Inference Attacks (MIA) (Shokri et al., 2017). Although we do not explore this setting in detail in this work, the proposed method can also be used for applications where complete unlearning is not desirable. This can be done by following a procedure similar to SCRUB+R (Kurmanji et al., 2023), wherein

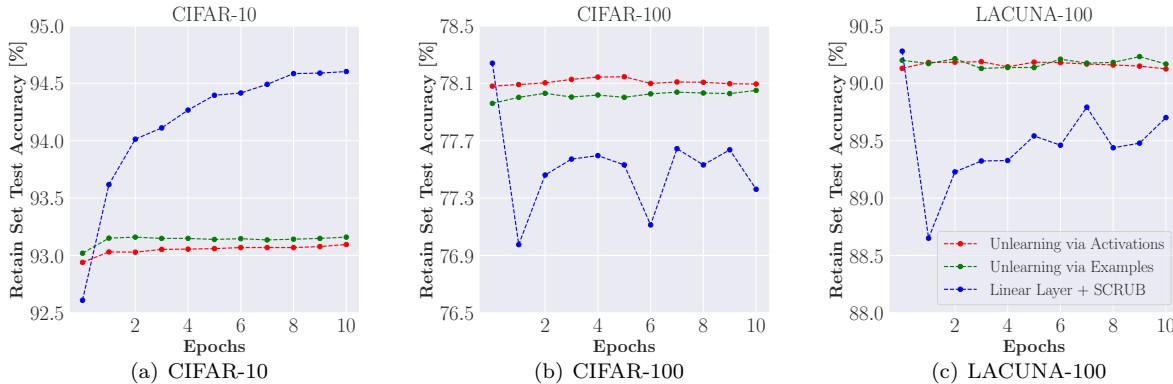

Figure 7: Comparison between the performance of proposed methods with added compute and the baseline on the retain set test data. For the proposed methods, the plots start from after the initial zero shot unlearning. For the baseline, the plots start from the original models. Retraining the models unlearned using the proposed models does not lead to any significant improvements in performance.

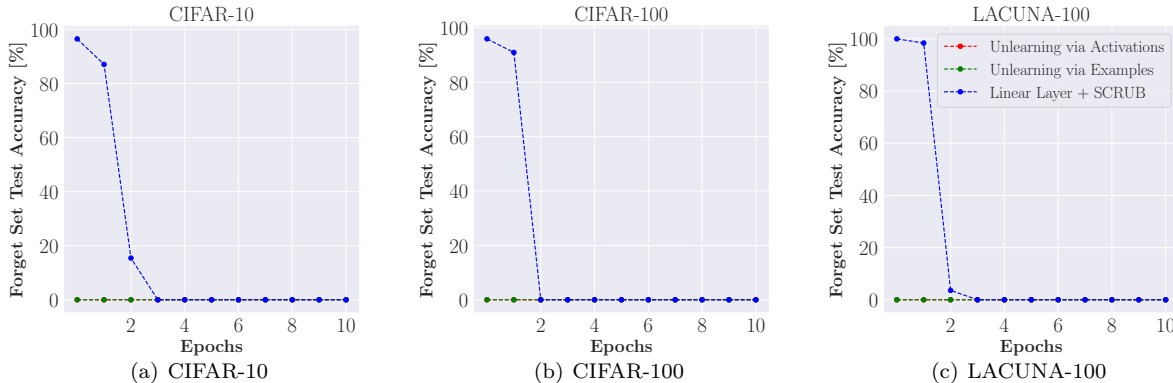

Figure 8: Comparison between the performance of proposed methods with added compute and the baseline on the and forget set test data. Note that for the proposed methods, the plots start from after the initial zero shot unlearning. For the baseline, the plots start from the original models. The green line occludes the red line since both of them stay at 0% throughout the training.

instead of selecting a particular model checkpoint, one can select the model corresponding to particular values of $N_a$ or $N_e$ such that the error on the forget set test data is similar to the reference point as defined in Kurmanji et al. (2023).

First,it is important to clarify that the proposed approaches are not a-priori suited for selective unlearning, i.e., the setting where we want the model to forget specific examples or a small subset of examples instead of removing the information about an entire class. The KV bottleneck essentially induces clusters of representation, where the members of a particular cluster correspond to the representations belonging to the same class (see Figure 2 in Träuble et al. (2023)). When we try to unlearn the representations corresponding to one particular example belonging to a particular class, the KV bottleneck routes the selection to other (key-)representations within the same cluster since those keys would be the next closest to the representation of the encoder. Since these representations also contain information about the same class as the examples we intend to unlearn, the model would still predict the class to be unlearned.

Due to the same reason the proposed approaches are also not designed for working against traditional Membership Inference attacks. According to the basic attacks setup as explained in Kurmanji et al. (2023),

the objective is to obtain a model that has unlearned a small subset of specific examples (i.e. selective unlearning) such that the loss of the model on the unlearned subset of examples should be indistinguishable from loss on examples that the model never saw during training.

Nevertheless, since the proposed approaches are designed for class unlearning specifically, we attempt to evaluate them on a modified version of the above. We call this "Class Membership Inference Attacks (CMIA)". In CMIA, the goal is to defend against an attacker whose aim is to determine whether a model that has undergone unlearning ever saw a particular class as a part of its training data. Thus, we want the model to unlearn a particular class such that the losses/performance of the model on the unlearned class as a whole, is indistinguishable from those on a held-out class that the model never saw during its training. We describe the experimental setup and results below.

**Experimental Setup** We perform the experiment for CIFAR-10 with a ViT/B-32 backbone. We divide the dataset into training data ($D_{Train}$), validation data ($D_{Val}$) and test data ($D_{Test}$). Training Data consists of 4000 examples per class; validation and test data consist of 1000 examples per class. We first trained a model on the first 9 classes of CIFAR10. Thus, class number 10 is the held-out class. Next, we unlearn class 1 from the model using the Unlearning via Activations approach introduced in the paper. We unlearn the model until the loss of the model on the validation sets of the forget class and the held-out class are similar. In our experiments, we find that we reach this point approximately at $N_a = 240000$. The loss $l(x, y)$ in our case is the cross-entropy loss.

Next, we label the losses corresponding to the validation and test set of the forget class as 1 and those corresponding to the validation and test set of the held-out class as 0. We train a binary classifier on the validation losses of the forget and held-out sets and evaluate it on the test losses. We follow a similar setting for the baseline model, where we obtain a model suitable for MIA defense by using SCRUB+R (Kurmanji et al., 2023). For a successful defense, we would want the accuracy of the classifier to be close to 50% on the test losses, indicating that it is unable to distinguish between the unlearned class and the held-out class. Same as Kurmanji et al. (2023), we use `sklearn.logistic_regression` as our attacker (the binary classifier). We call the approach described above Partial UvA (Partial Unlearning via Activations). We run experiments for 3 random seeds, and the mean of the attacker performance is reported. Note that a similar procedure can also be followed using Unlearning via Examples.

**Observations and Results**: We report the results of the experiment described above in the table given below. We observe that although the baseline performs slightly better, the proposed approaches perform competitively, even though we have not intended to develop the method for this scenario.

Table 10: Comparison on Class Membership Inference Attacks between the proposed approach and the baseline. A binary classifier is trained on the validation losses of the forget and held-out sets and is evaluated on the test losses. The proposed approach performs competitively to SCRUB + R

| Approach | Attacker Accuracy |
|---|---|
| Partial UvA | 53.50% |
| Linear Layer + SCRUB + R | **51.50%** |

### A.12 Mathematical and Algorithmic Formulations

In this section, we provide mathematical formulations for the proposed approaches of Unlearning via Activations and Unlearning via Examples as well as the empirical moving averages used for initializing the keys of the Discrete Key-Value bottleneck.

### A.12.1 Exponential Moving Averages for Key-Initialization

Similar to Träuble et al. (2023) we build upon exponential moving averages as introduced in Oord et al. (2017); Razavi et al. (2019). Below, we reiterate much of what is described in Träuble et al. (2023) (Appendix C). The set of equations given below describes the key initialization procedure. For each codebook $c$:

$$N_i^{(t)} := \gamma N_i^{(t-1)} + n_i^{(t)}(1-\gamma) \tag{1}$$

$$m_i^{(t)} := \gamma m_i^{(t-1)} + \sum_{j}^{n_i^{(t)}} E_{i,j}^{c(x)}(1-\gamma) \tag{2}$$

$$k_i^{(t)} := \frac{m_i^{(t)}}{N_i^{(t)}} \tag{3}$$

where $t$ is the index of the current mini-batch, $k_i$ and $N_i$ represent the position and counts of the $i$th key, $E^c(x)_{i,j=1\ldots n_i^{(t)}}^{(t)}$ are the $n_i^{(t)}$ head embeddings of the examples in the mini-batch which attach to the $i$-th key. We refer the reader to Appendix C of Träuble et al. (2023) for more details.

### A.12.2 Unlearning via Activations and Examples

In this section, we provide algorithmic implementations of the the proposed approaches of Unlearning via Activations (Algorithm 1) and Unlearning via Examples (Algorithm 2). Both algorithms are applied on model with a DKVB that was trained on the given task.

**Discrete Key-Value Bottleneck - Notations.** Before laying out the algorithms, we define some notations related to different components of the Discrete Key-Value Bottleneck (DKVB) (Träuble et al., 2023) that are essential for the algorithms. We refer the reader to Träuble et al. (2023) for the role of each of these components in the DKVB.

1. Pre-trained and frozen embedding model $E$

2. Random projection matrix $R$

3. Set of keys initialized using EMA $\{k_j\}_{j=0}^{N-1}$

4. Distance matrix $D \in \mathbb{R}^{|\mathcal{D}_{\text{train}}^{\text{forget}}| \times N}$, initialized to $-\infty$:

$$D[i,j] \leftarrow -\infty \quad \forall i \in \{0,1,2,\ldots,|\mathcal{D}_{\text{train}}^{\text{forget}}| - 1\}, \, j \in \{0,1,2,\ldots,N-1\}.$$

5. Selection mask $M \in \mathbb{R}^{|\mathcal{D}_{\text{train}}^{\text{forget}}| \times N}$, initialized to 1:

$$M[i,j] \leftarrow 1 \quad \forall i \in \{0,1,2,\ldots,|\mathcal{D}_{\text{train}}^{\text{forget}}| - 1\}, \, j \in \{0,1,2,\ldots,N-1\}.$$

We also define a function argsort as follows:

**Function Definition:** argsort
Let $v = [v_1, v_2, \ldots, v_n] \in \mathbb{R}^n$ be a one-dimensional array or vector. The function argsort$(v)$ returns a permutation $\pi$ of the indices $[1, 2, \ldots, n]$ such that:

$$v_{\pi(1)} \leq v_{\pi(2)} \leq \ldots \leq v_{\pi(n)}.$$

In other words, $\pi(i)$ is the index of the $i$-th smallest element of $v$.

For example, if $v = [3, 1, 4]$, then:

$$\text{argsort}(v) = [2, 1, 3],$$

since $v_2 = 1$ is the smallest, $v_1 = 3$ is the second smallest, and $v_3 = 4$ is the largest.

---

**Algorithm 1:** Unlearning via Activations

---

**Input:** Forget class training data $\mathcal{D}_{train}^{forget} \subset \mathcal{D}_{train}$, number of activations to be deactivated $N_a$, `top-k` parameter used for the DKVB

**Output:** Modified selection mask $M$

**Initialize:** Frequency matrix $f \in \mathbb{Z}_{\geq 0}^N$, initialized to 0:

$$f[j] \leftarrow 0 \quad \forall\, j \in \{0, 1, 2, ....., N-1\}.$$

**Step 1: Forward propagate the forget class training data through the model**
  **for** $i \leftarrow 0$ **to** $|\mathcal{D}_{train}^{forget}| - 1$ **do**

  $\quad x \leftarrow \mathcal{D}_{\text{train}}^{\text{forget}}[i]$

  $\quad e_x = R \cdot E(x)$

  $\quad$ **for** $j \leftarrow 0$ **to** $N - 1$ **do**
  $\quad\quad |\quad D[i,j] \leftarrow \|e_x - k_j\|_2 \times M[i,j]$
  $\quad$ **end**

  $\quad \mathcal{I}_e \leftarrow \text{argsort}(D[e_x, :])_{1:\texttt{top-k}}$

  $\quad$ **for** $j \in \mathcal{I}_e$ **do**
  $\quad\quad |\quad f[j] \leftarrow f[j] + 1$
  $\quad$ **end**

**end**

**Step 2: Deactivate the most frequently activated keys**
  $\mathcal{J} \leftarrow \text{argsort}(f)_{N - N_a + 1 : N}$
**for** $j \in \mathcal{J}$ **do**
  $\quad M[:, j] \leftarrow \infty$
**end**

---

## A.13 Training Details and Hyperparameters

We perform all of our experiments on a 48GB RTX8000 GPU. We do not use any data augmentation in any experiment. The transforms used for training the model with a ViT/B-32 backbone are the same as CLIP (Radford et al., 2021) pretrained ViT/B-32 transforms. For ResNet-50, both pre-trained weights and transforms are loaded from `torchvision.models.ResNet50_Weights`

### A.13.1 Training Details and Hyperparameters for training the original DKVB Models

In the case of ImageNet pretrained ResNet-50, the representations of the backbone are extract from the 3rd last layer for CIFAR-10, CIFAR-100 and LACUNA-100 and from the 4th last layer for ImageNet-1k. Table 11 shows all the hyperparameters used for training the base DKVB models.

### A.13.2 Training Details and Hyperparameters for training the original Baseline Models

For the baseline models, we deliberately train them to similar test ($\mathcal{D}_{test}$) accuracies as the models with a Discrete Key Value Bottleneck to ensure a fair comparison for unlearning. Table 12 shows the hyperparameters used for training the baseline models.

---

**Algorithm 2:** Unlearning via Examples

---

**Input:** Forget class training data $\mathcal{D}_{train}^{forget} \subset \mathcal{D}_{train}$, number of examples to be used for unlearning $N_e$, `top-k` parameter used for the DKVB

**Output:** Modified selection mask $M$ for the DKVB.

**Initialize:** Set of activated indices $\mathcal{I} \leftarrow \emptyset$

**Step 1: Randomly sample a subset $\mathcal{S}_f$ from $\mathcal{D}_{train}^{forget}$ of size $N_e$**

$$\mathcal{S}_f \sim \mathcal{D}_{\text{train}}^{\text{forget}}, \quad |\mathcal{S}_f| = N_e$$

**Step 2: Input the examples in the subset into the model to record the activated keys**

for $i \leftarrow 0$ to $|\mathcal{S}_f| - 1$ do

    $x \leftarrow \mathcal{S}_f[i]$

    $e_x = R \cdot E(x)$

    for $j \leftarrow 0$ to $N - 1$ do

        $D[i,j] \leftarrow \|e_x - k_j\|_2 \times M[i,j]$

    end

    $\mathcal{I} \leftarrow \mathcal{I} \cup \text{argsort}(D[i,:])_{1:\texttt{top-k}}$

end

**Step 3: Deactivate the activated keys**

for $i \in \mathcal{I}$ do

    $M[:,j] \leftarrow \infty$

end

---

Table 11: Hyperparameters used for training the base DKVB models

| Backbone | Hyperparameter | CIFAR-10 | CIFAR-100 | LACUNA-100 | ImageNet-1k |
|---|---|---|---|---|---|
| ViT/B-32 | top-k | 1 | 10 | 10 | 1 |
| | Key Dimension | 8 | 8 | 8 | 14 |
| | # of Key Init Epochs | 10 | 10 | 10 | 10 |
| | Type of Value Init | Gaussian Random | Zeros | Uniform Random | Zeros |
| | # of Codebooks | 256 | 256 | 256 | 256 |
| | # of Key-Value Pairs per Codebook | 4096 | 4096 | 4096 | 4096 |
| | Optimizer | Adam | Adam | Adam | Adam |
| | LR | 0.1 | 0.3 | 0.3 | 0.3 |
| | Batch Size | 256 | 256 | 256 | 256 |
| | Epochs | 74 | 71 | 7 | 3 |
| ResNet-50 | top-k | 1 | 2 | 1 | 1 |
| | Key Dimension | 14 | 14 | 8 | 14 |
| | # of Key Init Epochs | 10 | 10 | 10 | 10 |
| | Type of Value Init | Zeros | Random | Gaussian Random | Gaussian Random |
| | # of Codebooks | 256 | 256 | 256 | 256 |
| | # of Key-Value Pairs per Codebook | 4096 | 4096 | 4096 | 4096 |
| | Optimizer | Adam | Adam | Adam | Adam |
| | LR | 0.3 | 0.3 | 0.1 | 0.3 |
| | Batch Size | 256 | 256 | 256 | 256 |
| | Epochs | 70 | 4 | 1 | 5 |

### A.13.3 Training Details and Hyperparameters for SCRUB

For the baseline, we run SCRUB on the model with linear layer. One *epoch* consists of one *min step* and may or may not contain a *max step*. Hence the values of *min steps* and *epochs* are always same. One *max step* is included in every epoch for the first *msteps* epochs. We tune the hyperparameter *msteps* in our experiments and pick the case where the model is able to best recover its performance on the retain set test data and consider this model as the final unlearned model. We mention the hyperparameters used for running SCRUB corresponding to the results presented in Section 5.2.1 in Table 13. In this case, training of SCRUB is stopped when the forget set accuracy has either dropped to 0% or converged at a close to 0% value without damaging

Table 12: Hyperparameters used for training the baseline models

| Backbone | Hyperparameter | CIFAR-10 | CIFAR-100 | LACUNA-100 | ImageNet-1k |
|---|---|---|---|---|---|
| ViT/B-32 | LR | 0.001 | 0.01 | 0.01 | 0.01 |
| | Batch Size | 256 | 256 | 256 | 512 |
| | Epochs | 1 | 7 | 13 | 1 |
| ResNet-50 | LR | 0.01 | 0.001 | 0.01 | 0.001 |
| | Batch Size | 256 | 256 | 512 | 512 |
| | Epochs | 2 | 72 | 73 | 11 |

the retain set accuracy. Results presented in Appendix A.10 also use the same set of hyperparameters except *min-step* which is always 10 since we train all the methods for 10 epochs.

Table 13: Hyperparameters for SCRUB + Linear Layer Experiments shown in Section 5.2.1

| Backbone | Hyperparameter | CIFAR-10 | CIFAR-100 | LACUNA-100 | ImageNet-1k |
|---|---|---|---|---|---|
| ViT/B-32 | Forget Set Batch Size | 256 | 256 | 256 | 512 |
| | Retain Set Batch Size | 256 | 256 | 256 | 512 |
| | # of max-steps (*msteps*) | 3 | 9 | 5 | 3 |
| | # of min-steps / # of epochs | 3 | 10 | 7 | 3 |
| | LR | 0.001 | 0.01 | 0.01 | 0.001 |
| | Optimizer | Adam | Adam | Adam | Adam |
| ResNet-50 | Forget Set Batch Size | 256 | 256 | 256 | 512 |
| | Retain Set Batch Size | 256 | 256 | 256 | 512 |
| | # of max-steps (*msteps*) | 9 | 3 | 3 | 3 |
| | # of min-steps / # of epochs | 10 | 30 | 30 | 10 |
| | LR | 0.01 | 0.001 | 0.01 | 0.001 |
| | Optimizer | Adam | Adam | Adam | Adam |

### A.13.4 Training Details and Hyperparameters for Retraining Experiments

Once the DKVB models are unlearned using *Unlearning via Activations* and *Unlearning via Examples*, we retraining them in order to make a fair comparison with the baseline. Thus, during retraining, the initial performance of these models on the retain set is same as the final performance of the unlearned models. Table 14 show the hyperparameters used for retraining the unlearned DKVB models.

Table 14: Hyperparameters used for re-training experiments. **UvA** stands for *Unlearning via Activations* and **UvE** stands for *Unlearning via Examples*

| | CIFAR-10 | | CIFAR-100 | | LACUNA-100 | |
|---|---|---|---|---|---|---|
| | UvA | UvE | UvA | UvE | UvA | UvE |
| LR | 0.3 | 0.3 | 0.1 | 0.1 | 0.1 | 0.3 |
| Optimizer | Adam | Adam | Adam | Adam | Adam | Adam |
| Batch Size | 256 | 256 | 256 | 256 | 256 | 256 |
| Gradient Clipping | 0.1 | 0.1 | 0.1 | 0.1 | 0.1 | 0.1 |

