# OpenReview forum: "Low Compute Unlearning via Sparse Representations"
_TMLR — Accepted by TMLR_

### Review · Reviewer_wDTK · 2025-06-01

**Summary Of Contributions:**

The paper introduces two “compute-free” class-unlearning procedures: Unlearning via Activations and Unlearning via Examples, which exploit the sparsity of a DKVB inserted on top of a frozen pretrained encoder. This way, removing a class reduces to masking a few key–value pairs instead of running gradient updates. Across four image-classification benchmarks, the methods achieve successful forgetting of the target class while keeping the retain-class test accuracy almost unchanged and using much less compute than other baselines.

**Audience:**

Yes

**Broader Impact Concerns:**

In some cases, full class-level removal may be disproportionate and could harm model utility or fairness. I would suggest including a brief discussion on these implications in the paper.

**Claims And Evidence:**

No

**Requested Changes:**

1. I suggest including results using a fully trained linear-head baseline, in addition to the current early-stopped version.

2. Please discuss your choice of DKVB initialization (e.g., EMA) and how alternative strategies might affect performance.

3. I would appreciate a discussion addressing the setup concerns raised above.

**Strengths And Weaknesses:**

Strengths:

The proposed method is highly efficient, which leads to significant computational savings while maintaining competitive accuracy on retain classes. The method is conceptually simple, easy to implement, and validated across multiple datasets and backbones.


Weakness:

There are several concerns about the experimental setup and assumptions.
1. The evaluation splits the test set into forget and retain subsets. This is unusual, as the forget set is typically defined over the training set. While I understand this is for evaluation purposes, it only makes sense under class unlearning and assumes the training and test distributions are identical, which may not always hold in real-world scenarios. It would be better to acknowledge and discuss this limitation explicitly.

2. In Table 3, the linear-head baseline is deliberately undertrained so that its test accuracy matches the DKVB model. While this is presented as a “fair” comparison, I’m skeptical of its fairness as one model fully learns the training data, while the other does not. Since the comparison focuses on accuracy differences (rather than absolute accuracy numbers), I'm not sure why aligning test accuracy at the start is strictly necessary. Also, in practice, both models would be fully trained, and if introducing DKVB appears to reduce generalization slightly, then this trade-off should be reflected in the results.

3. The paper primarily compares unlearning performance relative to the original pre-unlearning model, rather than using the retrain-from-scratch oracle as the main reference. But in my opinion, the retrained model best represents the ideal post-unlearning behavior and should serve as the primary baseline. Additionally, the paper formalize the unlearning goal as “complete unlearning”, i.e. to achieve near-zero accuracy on the forget set, which may be valid for class unlearning in some cases but could be misleading in broader settings. For example, one can simply mask logits to remove a class without affecting remaining data. I think the goal should be matching the behavior of the retrained model, not simply destroying accuracy, particularly from a privacy and generalization standpoint.

4. The paper leaves open questions around DKVB initialization: why EMA was chosen, how other strategies compare, and how sensitive performance is to this choice. Moreover, the method has several prerequisites: access to the original training data or pre-cached activations, interventions to the training pipeline to record forget-class activations.

---

> ### Author Response · Authors · 2025-07-03
>
> We would like to thank the reviewer for carefully going through our paper and providing useful suggestions and feedback. We address the concerns and requested changes raised by the reviewer.
>
> The submission has been updated with all the changes requested by the reviewers (color coded in blue). We would be happy to engage in further discussion and incorporate further changes that the reviewer thinks might help improve the paper further.
>
> ## Forgetting over test set instead of the training set
> Typically practice is to define unlearning over the training examples instead of test examples. In our cases, we aim to analyse not only whether the the model has unlearnt "examples" of the forget class in the training data, we study  "unlearning" in a more general sense and wanted to analyse whether the model unlearns general "information" about the class, i.e., given a forget set, is the model's performance on all "similar" data points (and not just those belonging to the forget set); i.e. whether the unlearning generalizes. Hence we report the performance on the evaluation set. The performance on train set would always be destroyed since we use examples from the train set to select the activations to be discarded.
>
> Now, in our case, the unlearning will generalize as long as the OOD image (belonging to the same class) also corresponds to similar activations as the the examples in the forget set. In the case of the datasets used in the paper, this is true, presumable because the test data is similar to (i.e., not very "OOD" to) the train data and thus activates similar set of key codes. Whether the the unlearning will generalize to more OOD data depends on the OOD generalization abilities of the DKVB architecture. While studying the performance on more OOD images (belonging to the same forget class) is an interesting direction, we leave it for future work.
>
> We have edited the "Objective and Metrics" section in Section 5.1 to add the above discussion.
>
> ## Fully trained linear baseline
> Through our experiments, we aim to compare the proposed approaches with the baselines on two aspects
>
> (1) The compute required to unlearn the forget class
> (2) The damage incurred to the performance of the models on  retain classes upon unlearning the forget classes.
>
> The choice of training both the DKVB and the baseline models to similar accuracies was done so that unlearning on the baseline does not incur additional compute costs due its performance on the forget class being higher than that of the DKVB models (intuitively, higher the initial performance, more compute required for unlearning). Unlearning on DKVB could be incurring lower compute costs simply due to it lower initial performance. Thus, we wanted to eliminate this confounder in comparison.
>
> Nevertheless, the reviewer correctly points out that in practical settings both the models would be trained to completion. In Appendix A.6, we demonstrate that the proposed approaches perform equally well against baseline models trained to completion. The results show that the baselines when trained to completion are even more difficult to unlearn.
>
> ## Complete unlearning as the goal and comparison with the retraining baseline
>
> We set the objective to complete unlearning only to maximally _stress test_ the proposed approaches. Complete Unlearning would require highest values of $N_a$ and $N_e$. The higher the values of $N_a$ and $N_e$ the more representations corresponding to the forget set are deactivated leading to a higher likelihood of representations shared with the retain set being also deactivated, incuring maximal damage on the retain set. Thus, the damage incurred to the retain set during unlearning upto any extent is (on an average - since the representations to be deactivated are randomly selected) upper bounded by the damage incurred during complete unlearning.
>
> Note that complete unlearning is a more difficult objective than unlearning upto the extent of the retraining (in DKVB models) baseline. The observation that the proposed methods are able to achieve complete unlearning with minimal compute as well as minimal damage to the retain classes, implies that it would be able to achieve _unlearning upto the extent of the retraining baseline_ with even lower compute requirements and damage to retain class accuracy (given that the performance on the forget set in the retraining baseline would be greater than or equal to 0, the values for $N_a$ and $N_e$ would be less for unlearning upto the extent of the retrained model). Further, the performance of the model on the retain classes upon complete unlearning lower bounds (on an average) the performance on of the models on the retain classes after unlearning to any other arbitrary extent (including unlearning to the extent of the retraining baseline).
>
> We have modified the "Objective & Metrics" paragraph in Section 5.1 to include some of this discussion.

---

> > ### Author Response · Authors · 2025-07-03
> >
> > We would be happy to discuss any further concerns, arguments and  suggestions that the reviewer might have.
> >
> > ## Comparison with the baseline of deactivating the final logit of the linear layer
> >
> > We would like to argue that the proposed approaches are fundamentally as well as practically different (and more advantageous) than the trivial baseline of simply setting the class logits to zero (henceforth referred to as 'logit deactivation'). We would like the reviewer to consider the following points:
> >
> > * Logit Deactivation does not "remove" any information from the model. The information is still present in the weights of the linear layer. The proposed approaches on the other hand, are invasive on the (DKVB part of) the model and fundamentally destroy remove by deleting the corresponding representations.
> >
> > * While we set the objective to be "complete" unlearning in our experiments, the proposed approaches allow unlearning to _any_ extent by choosing the appropriate values of $N_a$ and $N_e$. This is not possible in Logit Deactivation. This is useful in cases where complete unlearning is not desirable.  We would like to refer the reviewer to Appendix A.10 for an example of such a requirement and the application of proposed approaches in the context of Membership Inference Attacks.
> >
> > ## Discussion about EMA
> > We choose EMA algorithm for initializing the keys codes of the DKVB, following Träuble et al. (2023). EMA (Razavi et al., 2019) is a standard approach for initializing codes in discrete bottlenecks such as VQ-VAEs (Van Den Oord et al., 2017). EMA in DKVB is important to ensure a clear separation of key codes corresponding to different classes.
> >
> > As requested by the reviewer, we have included new ablation experiments in Appendix A.4 demonstrating the importance of EMA based key initialization.
> >
> > We have also added a paragraph on Key-Initializations in Section 5.1 to include the above discussion
> >
> > [1] Träuble et al. (2023). Discrete Key-Value Bottleneck. https://arxiv.org/pdf/2207.11240
> > [2] Razavi et al. (2019). Generating diverse high-fidelity images with vq-vae-2. https://proceedings.neurips.cc/paper/2019/file/5f8e2fa1718d1bbcadf1cd9c7a54fb8c-Paper.pdf
> > [3] Van Den Oord et al. (2017). https://arxiv.org/abs/1711.00937
> >
> > ## Access to the original training data or pre-cached activations, interventions to the training pipeline to record forget-class activations
> > We believe that the reviewer might have mis-understood some of the pre-requisites of the proposed approaches.
> >
> > In standard unlearning settings where we have access to the forget class data, the proposed approaches do not require anything else in order to carry out unlearning. The examples of the forget class can be forward propagated through the models, the activations are recorded and subsequently discarded. The only requirement is having access to sufficient number of examples required to achieve unlearning upto the desired extent.
> >
> > Note that the proposed approaches do not require access to the retain class training data unlike baselines such as SCRUB. This make our approach a _zero-shot_ unlearning approach.
> >
> > Caching of activations would be useful in exceptional cases where we do not have access to even forget class examples at the time of unlearning. In such cases, since we already know which activations belong to which class, the activations corresponding to the forget class can be discarded to achieve unlearning.

---

> > > ### Author Response · Authors · 2025-07-09
> > >
> > > Dear reviewer,
> > > We would also like to refer you to our discussion with Reviewer AKz1 regarding the comparison of the logit deactivation baseline.

---

### Review · Reviewer_AKz1 · 2025-06-03

**Summary Of Contributions:**

The paper presents a method for unlearning classes from neural networks that have a DKVP layer. The approach is simple and very efficient. In experiments, the method is demonstrated on four benchmark datasets with two different computer vision backbones. The performance on retain set remains high throughout the unlearning process, and the performance is similar to other existing unlearning methods from the literature.

**Audience:**

Yes

**Claims And Evidence:**

Yes

**Requested Changes:**

- Key Initialization is unclear. Why do you need Dtrain for this? If you are encoding Dtrain, then presumably you also apply R1 (random projection matrix)?
- Typo?: "Ne examples are randomly sampled from the forget set training data (Dtrain^retrain)"
- This was unclear: "Therefore, on each dataset we select the class that is best learned by the respective
models with the Discrete Key Value Bottleneck trained previously, to be the forget class."
- The significance of the model reaching 0% accuracy is not immediately clear, so it would be helpful to have a sentence of explanation after, "The model unlearns the forget class completely between Na = 170, 000 (0.4%) and Na = 180, 000 (0%)."

**Strengths And Weaknesses:**

Strengths:
- Clear and well organized. Figure 1 is excellent!
- The method is simple and easy to implement.
- The method is super efficient (orders of magnitude faster than existing approaches).
- Experiments supported the conclusions and were presented well.

Weaknesses:
- The method requires the original model use a DKVP layer, so it it's not something that can be immediately applied to just any neural network. If one had an existing model, one would need to train a DKVP bottleneck on top of that backbone.
- The goal of unlearning was not well-defined, but I think this approach might not satisfy some of the desired unlearning criteria. The primary indicator of unlearning used in the experiments is keeping the retain set accuracy high while reducing the forget set accuracy to zero. But couldn't one do that with a trivial solution, e.g. by setting the logit of the forget class output neuron to negative infinity? The proposed approach is very similar to this trivial solution in that no weights are retrained.
- The reasoning wasn't compelling for using the best class as the forget class. I would think using random classes would be more straightforward.

---

> ### Author Response · Authors · 2025-07-03
>
> We would like to thank reviewer AkZ1 for going through our paper, asking insightful questions and suggesting useful changes. We attempt to address some of the concerns.
>
> The submission has been updated with all the changes requested by the reviewers (color coded in blue). We would be happy to engage in further discussion and incorporate further changes that the reviewer thinks might help improve the paper further.
>
> ## Approach similar to the trivial baseline of setting the class logit to infinity
> We would like to argue that the proposed approaches are fundamentally as well as practically different (and more advantageous) than the trivial baseline of simply setting the class logits to zero (henceforth referred to as 'logit deactivation'). We would like the reviewer to consider the following points:
>
> * Logit Deactivation does not "remove" any information from the model. The information is still present in the weights of the linear layer. The proposed approaches on the other hand, are invasive on the (DKVB part of) the model and fundamentally destroy remove by deleting the corresponding representations.
>
> * While we set the objective to be "complete" unlearning in our experiments, the proposed approaches allow unlearning to _any_ extent by choosing the appropriate values of $N_a$ and $N_e$. This is not possible in Logit Deactivation. This is useful in cases where complete unlearning is not desirable.  We would like to refer the reviewer to Appendix A.10 for an example of such a requirement in the context of Membership Inference Attacks.
>
> * We set the objective to complete unlearning only to maximally _stress test_ the proposed approaches. Complete Unlearning would require highest values of $N_a$ and $N_e$. The higher the values of $N_a$ and $N_e$ the more representations corresponding to the forget set are deactivated leading to a higher likelihood of representations shared with the retain set being also deactivated, incuring maximal damage on the retain set. Thus, the damage incurred to the retain set during unlearning upto any extent is (on an average - since the representations to be deactivated are randomly selected) upper bounded by the damage incurred during complete unlearning. We have modified the "Objective & Metrics" paragraph in Section 5.1 to clarify this.
>
> ## Key Initialization is unclear
> As discussed in Section 4 in the paper and Träuble et al. (2023), the purpose of key-initializations is to initialize the "key" vectors such that they cover the representation space of the pre-trained encoder sufficiently. We do key-initializations on $D_{train}$ to ensure that the key codes are best initialized with respect to the classes present in the training data (since the forget class would be one of the classes present in the training data). Initializing on $D_{train}$ ensures that keys corresponding to different classes present in the training data are well separated. The better the separation of keys corresponding to different classes, the fewer the representations (same as key-codes) shared between different classes and thus, better the unlearning quality.
>
> We have added an experiment in Appendix A.4 showing the effect of key-initializations on the final unlearning quality.
>
> The reviewer is correct in understanding that we apply the random projections R1 on the representations of the encoder. The representations are broken into several smaller heads, each which undergoes a random gaussian projection. This is done because the key-codes live in a smaller dimensional representation space than the representation of the encoder. This gives a combinatorial structure to the DKVB codebooks, allowing sparsity in updates as well as making the DKVB a robustness to distribution shifts, as discussed in Träuble et al. (2023).
>
> [1] Träuble et al. (2023). Discrete Key-Value Bottleneck. https://arxiv.org/pdf/2207.11240
>
> ## Typo
> We thank the reviewer for pointing out the typo. It should be  $D_{train}^{forget}. We have fixed the typo in the paper.
>
> ## Clarity regarding selection of the forget class
> We have re-written the "Unlearning" paragraph in Section 5.1 to explain this design choice. We hope that the modified writing is more clear.
>
> We would be happy to make any further changes that the reviewer thinks might help improve the clarity.
>
> ## Significance of complete unlearning
>
> We have added a small blurb elaborating the significance. We would be happy to further incorporate any particular discussion that the reviewer might have in mind.

---

> > ### Comment · Reviewer_AKz1 · 2025-07-04
> > **Logit deactivation is "shallow unlearning". How should we think of this method?**
> >
> > Thank you for the clear and detailed responses.
> >
> > I think the "logit deactivation" method is not as dissimilar as the authors suggest it to be in their response. Reviewer wDTK had the same thought as me, which suggests other readers will too. Here are some additional comments:
> > - I disagree with the response that logit deactivation does not remove information from the model --- it removes the vector of parameters associated with the output corresponding to the forgotten class. Thus, logit deactivation can be considered a form of "shallow" forgetting, in that it only modifies information in the output layer. It seems to me that the proposed method is also (somewhat) shallow, in that the modifications are limited to to the DKVB layer, but probably somewhere in between logit deactivation and "strong unlearning" in which information regarding the forget set is removed from every layer of a deep architecture. The authors touch on this in the Related Work Section, where they observe that the proposed methods affects intermediate rather than final model activations, while limiting the entanglement of features, but this discussion is vague.
> > - So how to characterize the difference between these methods? The criteria for "unlearning" that is provided --- getting poor accuracy on the forget set --- is too narrow to observe a difference. Presumably we want additional properties of the resulting network. These properties might depend on why we want to perform unlearning (E.G. privacy or freeing up capacity). The proposed method is superior on both these fronts compared to logit deactivation, but understanding and quantifying this would be useful.

---

> > > ### Author Response · Authors · 2025-07-07
> > >
> > > Thank you for elucidating your argument on the logit deactivation baseline. We misunderstood the logit deactivation baseline as simply setting the **one logit of the output vector** (occurring right before the softmax, of size C, where C is the number of classes) corresponding to the forget class as negative infinity instead of setting **the entire vector corresponding to the the forget class in the final linear layer** as negative infinity.
> > >
> > > We understand your concern about making the distinction with the logit deactivation baseline clearer and explicit. The distinction can be summarized in two points:
> > >
> > > * The proposed approaches allow control over the extent of unlearning, whereas logit deactivation does not. This has several advantages when it comes to unlearning objectives which are measured by metrics other than simply forget class test accuracy. One of such objective is Membership inference attacks.
> > > * From an interoperability perspective, the linear layer at the end of the network does not *extract / create* any new information about the class. Instead, its purpose is to perform a linear decision test on the information contained in the features of the penultimate layer. Intermediate layers (and their activations), on the other hand (like the ones contained in DKVB)  are responsible for more sophisticated information extraction from previous features, which actually affect the downstream classification of the data point. Thus, deactivating intermediate representations leads to destruction of information in a more concrete and "unrecoverable" manner as compared to simply deactivating a row of the final linear layer.
> > >
> > > In order to make the above discussion more explicit, we propose the following changes to the draft:
> > >
> > > * Add discussion related to the above points in the related works section. Contextualize the difference in terms of shallow vs deeper unlearning.
> > > * For quantifying the comparison, we can move the discussion about Class Membership Inference Attacks (CMIAs) in Appendix A.11 to the main paper and add the logit deactivation baseline to the comparison.
> > >
> > > If you deem the proposed changes appropriate and sufficient, we can go ahead and make these changes to the existing draft.

---

> > > > ### Comment · Reviewer_AKz1 · 2025-07-07
> > > >
> > > > I do not agree with the authors' characterization of the the difference between their method and logit deactivation, and I think clarity on this issue is important for fulfilling the claims and evidence requirement. I disagree on the distinctions they have made, and would like to see concrete distinctions made.
> > > >
> > > > 1.  The authors claim their method allows for control over the extent of unlearning, while logit deactivation does not. I disagree. One could perform a "partial logit deactivation" where the weights of the forget class output neuron are degraded (attenuated and/or have noise added), but not necessarily set to negative infinity. Adjusting the magnitude of the degradation would allow one to control the extent of unlearning.
> > > > 1. The authors claim in their rebuttal comment that the output layer "does not extract/create any new information about the class" but intermediate layers are "responsible for more sophisticated information extraction from previous features". I disagree. Mathematically, each layer activation contains a subset of the information in the previous layer, regardless of whether the layer is an intermediate or output layer.
> > > > 1. The authors claim that their method leads to "destruction of information in a more concrete and unrecoverable manner". This should be explained more clearly. The weights in the output layer are finely tuned, and relearning them after logit deactivation would requires access to the training data to relearn them (e.g. through linear regression). Can the authors say anything concrete to support the claim that relearning is harder in their method?

---

> > > > > ### Author Response · Authors · 2025-07-08
> > > > >
> > > > > Dear Reviewer,
> > > > > Thank you for further explaining your arguments. You raise several interesting points. We provide further discussion on some of the points you raised below and provide more concrete arguments for distinguishing the logit deactivation baseline from the proposed approaches.
> > > > >
> > > > > While from a mathematical point of view, each activation contains a subset of information from the previous activations, different layers extract different information. As we go deeper into the network, the information extracted by subsequent layers is more and more abstract. Eventually, the linear layer is responsible for learning the decision boundary over the features extracted by the penultimate layer.
> > > > >
> > > > > Consider the following: Depending on the complexity of the task, (for eg. prediction on more complex datasets,  image generation where the model involved a DKVB like discrete bottleneck, etc.) the DKVB would typically be placed at lower depths into the network. For eg. for training a DKVB based classification model on a more complex dataset (or more concretely, data on which the pretrained encoder has poor zero-shot performance), the model would look like "Pretrained & Frozen Encoder -> DKVB -> MLP -> Output" . In such cases, the DKVB would be responsible for extracting *relatively* lower level features as compared to the final linear layer. The linear layer, on the other hand, would simply be learning a linear decision boundary on the features refined by the previous layers (including those extracted by the DKVB if it exists in the model).
> > > > >
> > > > > To formulate it mathematically, a neural network could be represented as
> > > > >
> > > > > $o = h(\phi(x))$
> > > > >
> > > > > where $o$ is the output, $h(\cdot)$ represents the classifier on top of the features extracted by the feature extractor $\phi(\cdot)$. The DKVB (when present) would be considered a part of the feature extractor, whereas $h(\cdot)$ represents the linear layer. Now, if one destroys information in $h(\cdot)$ but leaves $\phi(\cdot)$ unchanged (i.e. the Logit Deactivation baselines), it would be easy to learn another $h(\cdot)$ on the features extracted by the feature extractor. However, if one destroys some information within the feature extractor (as in the proposed approaches), it is more difficult to relearn. Unlearning information in the feature extractor destroys.
> > > > >
> > > > > As an analogical example, it is easier to re-learn a linear classifier on top of a BERT pre-trained encoder as compared to recovering information unlearned from the BERT pre-trained encoder itself (due to reasons such as higher dimensionality, non-convex nature of the optimization, higher noise in lower-level layers, etc.).
> > > > >
> > > > > While this may not be apparent in the experiments discussed in the paper due to smaller scale, this becomes an important distinction at larger scales.

---

> > > > > > ### Comment · Reviewer_AKz1 · 2025-07-08
> > > > > >
> > > > > > Thanks. I agree with this, and would like to see it discussed somewhere in the text.
> > > > > >
> > > > > > To summarize, my issue is that the goal of unlearning is defined imprecisely in this paper. The paper uses the forget set accuracy metric as the only measure of success, and if this is the only metric, then logit deactivation is a superior method because it is simpler and easier to use. Thus, the authors need to explain that there are additional properties that we might want from unlearning. "Difficulty in relearning" is something concrete, though ideally this could be characterized more precisely.  The authors have proposed an efficient method to unlearn features at any layer of the network (as long as that layer is a DKVB layer), and that is a novel contribution that helps achieve the goal of difficulty in relearning. The experiments in this paper do not explore the performance of unlearning in DKVB layers at different levels of the network, but this is something that could be left for future work.

---

> > > > > > > ### Author Response · Authors · 2025-07-09
> > > > > > >
> > > > > > > Thank you for summarizing your arguments. We will update the current draft with discussion on this topic shortly.
> > > > > > > We would also like to thank you for engaging in this insightful discussion with us and providing useful feedback.

---

> > > > > > > > ### Author Response · Authors · 2025-07-11
> > > > > > > >
> > > > > > > > Dear reviewer,
> > > > > > > > We have updated the limitations sections in the draft with the discussion above.

---

> > > > > > > > > ### Comment · Reviewer_AKz1 · 2025-07-11
> > > > > > > > >
> > > > > > > > > The latest version has grammatical errors in the limitations discussion. Please check.

---

> > > > > > > > > > ### Author Response · Authors · 2025-07-11
> > > > > > > > > >
> > > > > > > > > > We have fixed the errors. Thank you for pointing it out.

---

### Review · Reviewer_2ZFu · 2025-06-25

**Summary Of Contributions:**

This paper studies how to effciently do machine unlearning. Existing works often require extensive computation to forget the forget set while keeping the performance on the retain set. This paper utilizes the characteristics of sparse representations to achieve low computation in machine unlearning with negligible damage to the model performance on retain set. Experimental results are provided to support its claims.

**Audience:**

Yes

**Claims And Evidence:**

Yes

**Requested Changes:**

* It would be good to update the related work section with recent studies.
* It would be good to incorporate some clock time comparison with baseline methods.
* It would be good to conduct some ablation studies.

**Strengths And Weaknesses:**

Strengths:
* The paper is well-organized and easy to follow, with a clear motivation.
* The topic is important, and the proposed method is practical, which would contribute to the deployment of the unlearning method in real-world tasks.
* It provides good insights about the relationship between unlearning and sparsity, which could be beneficial to the related field.
* Experiments on different data and architectures are provided and demonstrate the improvements.

Weaknesses:
* Most of the related works are from or before 2023. It would be better to also incorporate more recent studies in this field.
* Apart from FLOPs, it would be better to include more results about the clock time comparison with baseline methods to better support the claim in the paper.
* More ablation studies would be good for a better understanding of the proposed methods.
* In more general scenarios, given an arbitrary model with a forget set, I was wondering if your method or some variant of your method can still work with low compute. This would make the proposed method useful in more scenarios.

---

> ### Author Response · Authors · 2025-07-03
>
> We thank the reviewer 2ZFu going through our paper and providing useful feedback and suggesting changes that would improve the paper. Below, we address some of the reviewers comments.
>
> The submission has been updated with all the changes requested by the reviewers (color coded in blue). We would be happy to engage in further discussion and incorporate further changes that the reviewer thinks might help improve the paper further.
>
> ## More related work
> Thank you for pointing this out. There have indeed been some interesting studies in the recent years on unlearning. We have updated the related works section to discuss the relevant ones.
>
> We would be happy to include discussion about any particular works that the reviewer might have in mind.
>
> ## Clock times as a proxy for compute efficiency
> Runtimes of different algorithms is an important factor affecting the feasibility of using an unlearning algorithm, especially in the case of systems in production. In order to highlight the runtime efficiency of the proposed algorithms, we have updated the paper with a comparison of the runtimes required for the unlearning procedure done via the proposed approaches versus via SCRUB, our strongest baseline. The comparison is discussed in Appendix A.5, with a reference in Section 5.3. The comparison shows that the proposed approaches are atleast 20 times faster than SCRUB.
>
> ## Further ablation studies
> We have updated the paper with the following ablation studies
>
> * Ablation study studying the influence of key-initializations on the final unlearning quality in Appendix A.4
> * Updated Appendix A.8 to include a comparison with SCRUB in the case of unlearning with the forget class chosen randomly
> * Included a comparison with the baselines on linear layer + backbone models that were trained to completion (instead of baseline models that were trained to accuracies similar to the DKVB models) in Appendix A.9
>
>
> ## Application of Method or its variants to arbitrarty models
> We thank the reviewer for raising this interesting question.
>
> The applicability of our method boils down to the existence of a DKVB-like bottleneck in the said model which induces class-wise separation of representations which allows discrading representations corresponding to a subset of classes without significantly affecting the performance on other classes. While the adoption of DKVB-style bottlenecks in general purpose models is an interesting future direction, we attempt to give the reviewer a sense how potential extensions of our work might look like.
>
> Many widely used vision models (for eg. DALL-E, stable diffusion, etc.) already use some form of discrete-bottlenecks. An interesting direction of research could involve investigating how these bottleneck can be replaced by DKVB style bottlenecks. Additionally, there is also evidence of successful adoption of DKVB in langauge models [1].
>
> Generally, as long as the keys are well separated, the training objective used to train the values of the DKVB and the following decoder are very flexible. Typical training objectives for general purpose models and transferrable representations such as contrastive losses or next token prediction can be easily used in the case of models with DKVB as well.
>
> Is it possible to unlearn arbitrary forget sets: For unlearning, it would be important to have an idea of what kind of concepts (or in a more general sense "behaviors") might be desired to be unlearnt. These concepts / behaviors are analogous to classes in our cases. The basic requirement is that the constituents of the forget set should have a shared underlying structure (for eg. the classes in CIFAR datasets represent a common "object"). Once this requirement is met, auxiliary objectives (such as contrastive losses) could be then used to train DKVBs (which could be a part of the larger model) to contain  representations encoding these concepts which are well separated from each other. Procedures similar to UvA and UvE could then be employed in order to unlearn these behaviors or concepts.
>
> [1] Diera et al., 2024. Continual Learning for Encoder-only Language Models via a Discrete Key-Value Bottleneck. https://arxiv.org/abs/2412.08528

---

### Decision · Action_Editor_49eM · 2025-08-04

**Recommendation:** Accept as is

**Audience:**

Yes

**Audience Explanation:**

The paper introduces a novel approach for unlearning in pre-trained models, given a designated forget set. The objective is to remove specific knowledge from the model while preserving its performance on the remaining data.

Reviewers generally agreed that the proposed method is both intuitive and efficient and recognized the paper as a meaningful contribution to the field of machine unlearning. Some concerns were raised, particularly regarding the evaluation methodology. However, these might reflect broader challenges in the field rather than flaws specific to the paper. Additionally, two reviewers pointed out that a simple baseline—zeroing out logits for classes in the forget set—might also perform well under the paper's evaluation metric. The authors addressed this point in the rebuttal by adding a discussion to the revised draft. While further exploration of this baseline is warranted in future work, the current paper already offers a valuable and well-substantiated contribution. I therefore recommend acceptance.

**Claims And Evidence:**

Yes

**Claims Explanation:**

The main claim of the paper is that the proposed approach “efficiently unlearns the forget set and incurs negligible damage to the model’s performance on the rest of the dataset.” The paper evaluates unlearning performance by measuring the drop in accuracy on the forget set and presents experimental evidence across four image classification benchmarks supporting this claim. Some reviewers noted limitations in the evaluation metric itself and highlighted the need for more nuanced metrics in future work, but I don't think this invalidates the claims of the paper.